# Topotaxially grown composite cathodes for cobalt-free high-energy long-life Li-ion batteries

Junyi Yao[1,11,12], Sizhan Liu[2,12], Wujun Zhang[3], Lijun Wu[4], Zhenjie Zhang[5], Ping He[5], Yanbin Shen[3], Liwei Chen[3,10]✉, Mingyuan Ge[6], Lu Ma[6], Xiaotian Zhu[1], Kaihua Xu[7], Kun Zhang[7], Feng Wang[8], Jianqing Zhao[1,9]✉ & Jianming Bai[6]✉

The vehicle industry's increasing demand for electrification necessitates the removal of expensive and rare cobalt from current high-energy batteries. However, eliminating cobalt poses challenges due to its vital role in maintaining the layered structural ordering and cycling stability of commonly used $Li(NiMnCo)O_2$ cathodes. As an alternative to conventional layered oxide designs, we report a lithium nickelate cathode with a composite structure comprising major stoichiometric layered and minor rocksalt phases within the same oxygen lattice. This material outperforms conventional designs by maintaining stable battery operation at voltages up to 4.8 V vs. Li|Li$^+$, with 88% capacity retention after 1000 cycles at 2C. The topotaxial-growth-enabled interlock between the two components mitigates chemo-mechanical degradation, offering a promising pathway to cobalt-free cathodes. Additionally, we reveal a miscibility gap in the Li-Ni-O system that enables kinetic adjustment of composition and structure during sintering, thereby tuning the functionality of high-energy cathodes.

The electrification of vehicle industry is critically dependent on high-energy, long-life, and safe Li-ion batteries (LIBs). Among various cathodes for LIBs powering electric vehicles (EVs), the layered Ni-Mn-Co oxides (NMC) with high Ni contents are the most promising due to their high capacity and low cost[1,2]. However, high-Ni NMC cathodes have several pitfalls, including difficulties in synthetic control of the layering, along with their poor cyclability, which are generally addressed through optimizing the composition and processing conditions, aiming to obtain controlled bulk and surface compositions that mitigate capacity fade. $LiNiO_2$ is an archetypal system within the large NMC composition space, discovered as early as the 1950s[3], but has only recently been considered a cathode material due to the growing interest in developing cobalt-free cathodes[4]. However, cycling instability remains the most serious problem preventing $LiNiO_2$ from being used as a practical cathode material. Several approaches have been explored to improve the cycling stability of $LiNiO_2$ and other high-Ni NMC cathodes in recent years, such as the introduction of minor defects at the Ni sites to

[1]College of Energy, Jiangsu Key Laboratory of Advanced Negative Carbon Technologies, Soochow University, Suzhou, PR China. [2]Interdisciplinary Science Department, Brookhaven National Laboratory, Upton, NY, USA. [3]i-Lab, Suzhou Institute of Nano-Tech and Nano-Bionics (SINANO), Chinese Academy of Sciences, Suzhou, PR China. [4]Condensed Matter Physics and Materials Science Division, Brookhaven National Laboratory, Upton, NY, USA. [5]Center of Energy Storage Materials & Technology, College of Engineering and Applied Sciences, National Laboratory of Solid State Microstructures, Nanjing University, Nanjing, PR China. [6]National Synchrotron Light Source II, Brookhaven National Laboratory, Upton, NY, USA. [7]GEM Co., Ltd., Shenzhen, PR China. [8]Applied Materials Division, Argonne National Laboratory, Lemont, IL, USA. [9]Jiangsu Zoolnasm Technology Co., LTD, Suzhou, PR China. [10]School of Chemistry and Chemical Engineering, Shanghai Jiao Tong University, Shanghai, PR China. [11]Present address: Department of Chemistry, Virginia Tech University, Blacksburg, VA, USA. [12]These authors contributed equally: Junyi Yao, Sizhan Liu. ✉e-mail: lwchen2018@sjtu.edu.cn; jqzhao@suda.edu.cn; jmbai@bnl.gov

                                                  

suppress structural distortion[5], tungsten doping to protect the surfaces of primary particles[6], and an in-situ process to generate a stable fluoride- and boron-rich cathode–electrolyte interface to protect the cathode from parasitic reactions with the electrolyte[7].

Structural engineering, employing a dual-phase composite with coherently integrated domain boundaries[8–10], has proven to be an effective way to ameliorate anisotropic volume variation during electrochemical cycling, thereby addressing the cycling instability of high-Ni NMC cathodes. The synergy between the two constituent phases depends on their relative electroactivity, where the electroactive component contributes to the specific energy, and the other mitigates the lattice distortion through the coherent domain interface to enhance cycling stability. Most such heterostructures developed so far were created by introducing foreign elements via cationic substitution or doping in the pristine layered structure, as thermodynamically directed end-products[8,11–14]. The doping procedure poses difficulties in scale-up production and may cause undesirable defects in the host structure. It is advantageous—but also a great challenge—to construct a dual-phase heterostructure to stabilize a pure Li-Ni-O system.

In this work, we report a kinetic control strategy to construct an interlocked dual-phase composite by tuning the integrated structures and their phase ratio in the Li-Ni-O system. The two phases in the composite, a layered $LiNiO_2$ (space group $R\bar{3}m$) and a rocksalt $Li_xNi_{2-x}O_2$ (space group $Fm\bar{3}m$), form a quasi-binary section in the ternary Li-Ni-O system (Supplementary Information, Fig. S1), and are topotaxially related throughout the solid-state synthesis. Using in-situ synchrotron X-ray diffraction (SXRD) measurements, we show that the biphasic process proceeds within a considerably wide temperature and time window, making it feasible to generate a kinetically stable dual-phase composite with desirable functionality. Based on this finding, we successfully synthesized an interlocked dual-phase composite $(LiNiO_2)_L|(Li_xNi_{2-x}O_2)_{RS}$ (L and RS represent layered and rocksalt phases, respectively) cathode to address the long-standing cycling instability issue in layered cathodes. The intrinsically interlocked layered and rocksalt domains within an individual primary particle demonstrate enhanced structural stability countering cycling-induced chemo-mechanical degradation, as verified by *operando* SXRD measurements and ex-situ transmission X-ray microscopy (TXM) imaging. The substantial mitigation of intercalation-induced anisotropic lattice distortion enables stable battery operation under a high cutoff voltage of 4.8 V vs. Li|Li+ with an 88% capacity retention after 1000 cycles at a 2 C charge/discharge rate.

## Results and discussion

### The interlocked rocksalt-layered composite originated from topotaxial crystal growth

The overall process for the synthesis of $LiNiO_2$ from hydroxides is known to proceed from the initial $Ni(OH)_2$ and $LiOH$ hydroxides to a partially-lithiated rocksalt $Li_xNi_{2-x}O_2$ phase, and then to the final layered $LiNiO_2$ phase[15]. In this work we will focus on exploring the rocksalt to layered phase transition and the interlock relationship between the two structures. Of particular interest is that the rocksalt-to-layered transformation does not necessarily follow the solid solution phase transition route indicated in the early studies[15,16]. At moderate sintering temperatures, due to the sluggish Li-ion transport, the non-equilibrium reaction proceeds as a quasi-binary phase transition, forming a metastable dual-phase structure intergrown within a single primary particle. This kinetically dominated reaction route, as revealed by our in-situ SXRD analysis, will be discussed in more detail in the following sections. The crystallographic analysis of the dual-phase structure presented several complexities, as explained in the 'Notes on the SXRD Data Analysis,' in the Supplementary Information, which includes details on the Rietveld refinement-based structural and quantitative analysis during the sintering process of the Li-Ni-O system. Figure 1a shows the SXRD pattern of the $(LiNiO_2)_L|(Li_xNi_{2-x}O_2)_{RS}$

composite synthesized by sintering the hydroxide precursors at 600 °C for 6 h (600C-6h), with Rietveld refinements used to quantify the phase fraction and structural parameters (Supplementary Information, Table S1). The SXRD analysis demonstrates that the 600C-6h sample assumes a dual-phase composition of the rocksalt $Li_xNi_{2-x}O_2$ and the layered $LiNiO_2$ phases with a mass ratio of 13:87, rather than a single layered phase with pronounced Ni mixing in the Li layers[17]. The dual-phase composite exhibits a miscibility gap in the Li-Ni-O system, characterized by two distinct phases with different structures and lithium occupancies. This finding stands in stark contrast to the conventional polymorphic phase transition model, which assumes similar lithium contents in the rocksalt and layered phases. This insight was instrumental in enabling the synthesis of the dual-phase composite, with tuned composition and structural properties for use as a high-performance cathode material in LIBs. The SXRD analysis suggests a topotaxial relation between the rocksalt and the layered phase, specifically, the layered phase nucleates and grows within the rocksalt crystal, with its layer order develops along one of the four cubic <111>$_{RS}$ directions (Fig. 1b) of the rocksalt host[15]. To confirm the intergrowth nature of the dual-phase composite, we conducted electron microscopy measurements to provide direct evidence for the composite structural model.

Domains of the two component crystals, i.e., the rocksalt $Li_xNi_{2-x}O_2$ and the layered $LiNiO_2$, each with different lattice orientations, are both captured in the cross-sectional HAADF-STEM image taken from the same composite sample investigated in the SXRD analysis. In Fig. 1c, three crystal domains—one rocksalt (d1) and two layered phases (d2 and d3), with their (001) planes parallel to the RS (111) and ($\bar{1}$11) planes, respectively—are identified. The 2D fast Fourier transform images in the side panel, taken from the three selected regions, corroborate this identification. As evidenced by the HAADF-STEM image, the rocksalt $Li_xNi_{2-x}O_2$ and the layered $LiNiO_2$ domains share the same cubic close-packed lattice, with coherent grain boundaries formed via the topotaxial growth mechanism. A typical $LiNiO_2|Li_xNi_{2-x}O_2$ grain boundary is shown in Fig. 1e, a magnified image from the purple rectangle in Fig. 1c. While the Li and O atoms are not visible, the Ni columns and the Ni/Li columns are clearly visualized as arrays of strong and weak spots. This shows that the rocksalt ($\bar{1}$11)$_{RS}$ plane and the layered (001)$_{L2}$ plane are well aligned across the boundary, characterized by a low coincidence lattice mismatch:

$$\delta = \frac{d_{L003} - 2d_{RS111}}{2d_{RS111}} = -0.0053$$

The d-spacings are calculated based on the lattice parameters of the rocksalt and layered phases obtained from SXRD analysis (Table S2). This low lattice mismatch indicates a strong elastic coupling between the two components in the dual-phase composite. In other words, the topotaxially grown layered and rocksalt domains are inherently interlocked within individual primary particles, playing a critical role in stabilizing the cathode during electrochemical cycles. Further discussions on this topic is provided in later sections.

Even though the layered crystal nucleates with equal probability along the four <111>$_{RS}$ orientations, only one direction is adopted in each isolated domain to minimize the free energy[15]. As the layered crystal domains nucleated at different sites grow, they may border each other and form different kinds of grain boundaries. Figure 1f shows a coherent twin boundary formed by two layered domains, L1 (along [111]$_{RS}$) and L2 (along [$\bar{1}$11]$_{RS}$). The dashed gray line (Supplementary Information, Fig. S9) indicates the coincidence site lattice, or the mirror plane, between the twins. The intersection angle between L1 and L2 is 70.53°, which corresponds to the angle between [111]$_{RS}$ and [$\bar{1}$11]$_{RS}$ (Fig. 1b), in accordance with the topotaxial rocksalt-to-layered phase growth mechanism. Because of this high angle grain boundary, the [001]$_{L1}$ vector has a large projection in the a–b planes of its twin

                                                                 

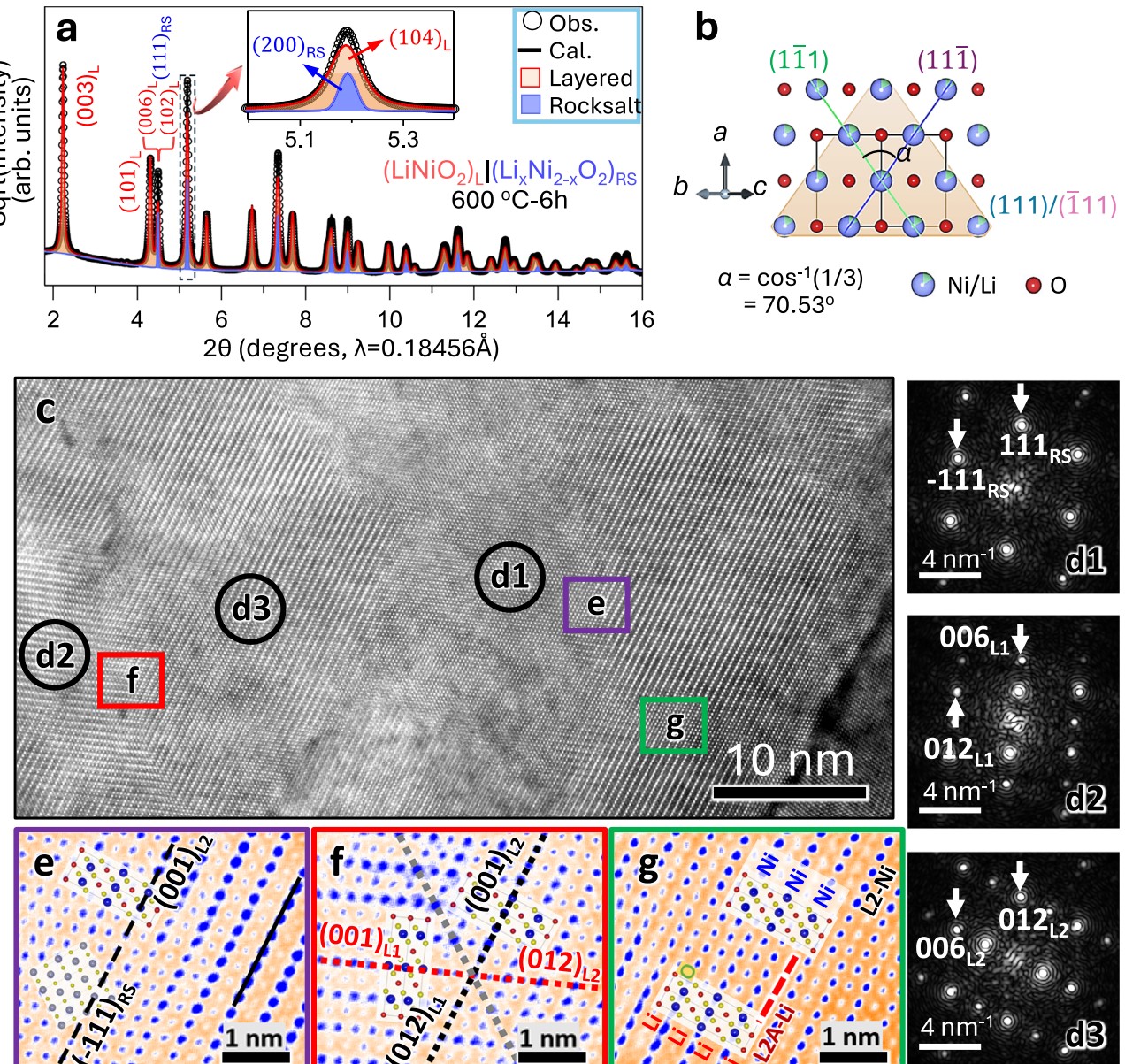

**Fig. 1 | $(LiNiO_2)_L | (Li_xNi_{2-x}O_2)_{RS}$ composite cathode. a** SXRD pattern of the 600C-6h sample showing a composite structure with 87 wt% layered (L; red) and 13 wt% rocksalt (RS; blue) phases. The inset shows the overlap of the $104_L$ and $200_{RS}$ peaks. A full-scale plot of the SXRD pattern with refinement information is provided in Fig. S7. **b** Schematic illustration of the rocksalt structure of a Li-containing nickel oxide with its four cubic $<111>_{RS}$ vectors as accessible directions to initialize the layer ordering in the topotaxial transformation. **c** Cross-sectional HAADF-STEM image (scale bar = 10 nm) of the $(LiNiO_2)_L|(Li_xNi_{2-x}O_2)_{RS}$ composite obtained by sintering at 600 °C for 6 h. **d** Fast Fourier transform (FFT) patterns (scale bars = 4 nm⁻¹) from the area marked by circles labeled d1, d2 and d3, respectively, showing RS (d1) and two layered orientation domains L1 (d2) and L2 (d3) in the composite. Magnified images (scale bars = 1 nm) from areas marked by (**e**) purple, (**f**) red and (**g**) green rectangles in (**c**), showing boundaries between (**e**) RS and layered domain L2, (**f**) layered domains L1 and L2, and (**g**) layered domain L2 and its antiphase domain L2A. The images are shown in false color for clarity. The lines in (**e**–**g**) mark the labeled planes (the planes are perpendicular to the screen). The layered structure model is embedded in (**f**) and (**g**), respectively. Here, L1 and L2 represent the two layer-structured domains with their (001) planes parallel to RS (111) and ($\bar{1}$11) planes, respectively, e.g., $(001)_{L1}||(111)_{RS}$ and $(001)_{L2}||(\bar{1}11)_{RS}$, as shown in (**d**). A colored version of the HAADF-STEM image showing the phase boundaries, and enlarged views of e, f and g are provided in Supplementary Information as Figs. S7 and S8, respectively. Source data for a are provided as a Source Data file.

crystal L2 across the boundary. This generates a coupling between $a$ and $c$ lattice parameters, which also helps stabilize the cathode during electrochemical cycling. More details about the $a$–$c$ coupling are provided in Supplementary Information (Fig. S9).

When two growing layered domains of the same crystal orientation border each other, they may merge into a single crystal domain through parallel growth or, with an equal probability, they may grow into two crystals with an antiphase boundary (APB) instead. Figure 1g shows such a grain boundary, where L2 and L2A domains meet with opposite cation ordering, while with their anionic sublattice remains the same. APBs have been reported to be present in pristine layer-structured cathodes and tend to propagate during electrochemical cycling, accompanied by the formation of other defects, such as cation dislocations and oxygen vacancies, which reduce the Li-ion mobility and capacity retention over prolonged cycles[18]. Similarly, APBs can also form between two layered domains with different crystal orientation, forming asymmetric twin crystals, and have a similar negative impact on the cathode stability[19].

In our 600C-6h composite, as mentioned earlier, there is 13.0 wt% rocksalt phase, which may effectively separate most layer-structured domains from direct contact. In Fig. 1g, the two antiphase domains are

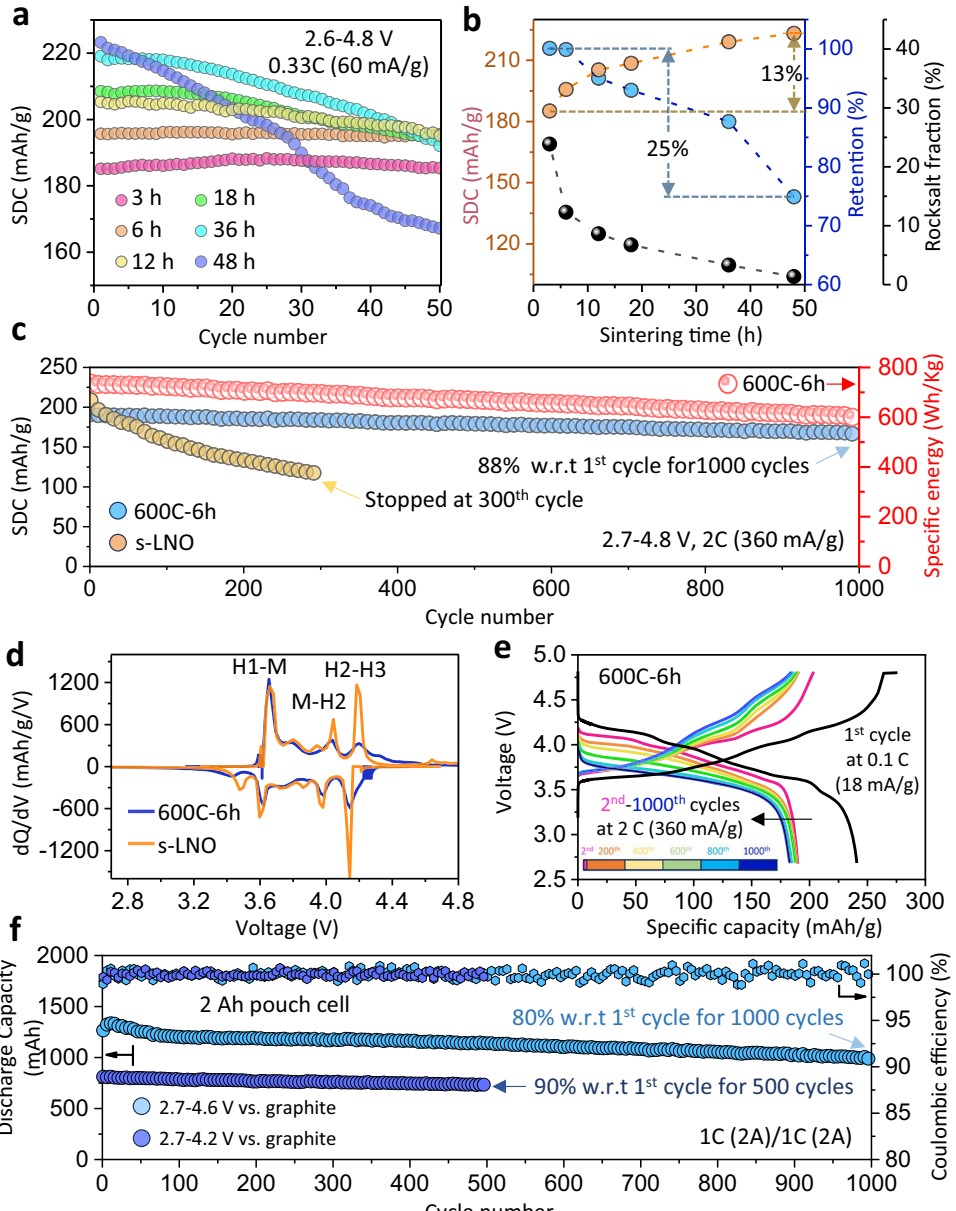

**Fig. 2 | Electrochemical performance of composite cathodes. a** Specific discharge capacity (SDC) of 600C-xh cathodes, with varied sintering duration x, from 3 to 48 h, at 0.33 C in a voltage range of 2.7–4.6 V vs. Li|Li⁺. **b** Initial capacities, capacity retentions (after 50 cycles) and the rocksalt fraction as a function of sintering duration (x) for 600C-xh cathodes. **c** Comparison of the cycling performance between the dual-phase 600C-6h and the single-phase s-LNO cathodes at 2 C in an extended voltage range of 2.7–4.8 V vs. Li|Li⁺ over 1000 cycles, along with

the corresponding specific energy of the 600C-6h cathode calculated from delivered capacity and average working voltage. **d** Differential capacity (dQ/dV) profiles of the two cathodes recorded at 0.1 C during the first cycle. **e** Charge/discharge curves of the 600C-6h cathode for the initial cycle at 0.1 C and subsequent cycles a 2 C, up to the 1000th cycle. **f** Cycling performance of 2 Ah 600C-6h||graphite pouch cells at 1 C in voltage ranges of 2.7–4.2 and 2.7–4.6 V, respectively. Source data for a, b, c, d, e, and f are provided as a Source Data file.

partially divided by a narrow area of residual rocksalt phase. It is also possible that some rocksalt-like phases are generated when two anti-phase domains come into contact during the sintering process. Due to potential APB formation upon direct contact between layer-structured domains, we infer that in a dual-phase lithium nickelate composite with good cycling stability must contain a sufficient amount of rocksalt component, such that the coherent LiNiO₂|Li$_x$Ni$_{2-x}$O₂ grain boundaries dominate the interface regions. The inference is further examined in the next section.

## Tuning of the dual-phase structure for optimal performance

All existing structural or morphological modification strategies for layered NMC cathodes to improve their long-term electrochemical

performance involve a trade-off between specific capacity and cycling stability[20]. The key to achieving optimal cyclability with minimal capacity sacrifice in a dual-phase composite cathode lies in balancing the amount of structural stabilizer—i.e., the rocksalt phase—with the active layered component. Figure 2a compares the cycling performance of different (LiNiO₂)$_L$|(Li$_x$Ni$_{2-x}$O₂)$_{RS}$ composite cathodes in Li|| LNO cells at 0.33 C (1 C corresponds to a current density of 180 m Ag⁻¹) in the voltage range of 2.7–4.6 V vs. Li|Li⁺ after an electrochemical activation at 0.1 C for three cycles. The samples were synthesized by heating the hydroxide precursors at 600 °C for increasing durations (denoted as 600C-xh, where x represents sintering hours). As shown in Fig. 2a, the samples that sintered for 3 and 6 h show the most stable cycling behavior, with no signs of capacity decay after 50 cycles.

Notably, 600C-6h sample delivers an initial capacity of 195.7 mAhg$^{-1}$ and outperforms the others in both cycling stability and specific capacity after 50 cycles. Figure 2b shows that the rocksalt phase fraction in the 600C-xh composite cathodes is kinetically adjustable based on sintering hours (Table S1). Higher rocksalt content correlates with reduced initial capacity but enhanced cycling stability. The sensitivity of this kinetic control is highlighted by a 25% variation in capacity retention and a 13% change in initial discharge capacity, with the rocksalt fraction ranging from 2.2 to 20.1 wt%. These findings support the inference that a higher rocksalt phase ratio promotes the formation of coherent boundaries between layered and rocksalt domains, thereby enhancing cathode stability during electrochemical cycling.

In addition to examining the kinetic evolution of the structural and morphological parameters of the 600C-xh samples (Table S1), we also studied the kinetic effects on the secondary particles of the 600C-xh cathodes. As shown by the SEM images (Fig. S10), the shape and size of the secondary particles remain largely unchanged, while internal porosity shows a slight increase with longer heating durations at 600 °C (Fig. S11). Based on prior studies on the influence of particle/crystal size[21,22] and internal porosity[23–25] on cycling stability, we confirm that the rocksalt phase fraction is the most critical factor governing the long-term performance of the dual-phase composite cathode.

A detailed electrochemical evaluation of 600C-xh cathodes with varying sintering times was conducted, with the main results presented in Fig. S12. The charge and discharge curves during the first cycle show progressively increasing capacities with longer sintering times, which can be attributed to the increasing fraction of the layered phase serving as the active cathode component (Fig. 2b).

Interestingly, cathodes with longer sintering times exhibit higher polarization, manifested by steeper voltage slopes at high state of charge (SOC), compared to those sintered for shorter durations, such as the 600C-6h cathode. This behavior is an intrinsic property of single-phase LNO cathodes, where severe c-lattice shrinkage induced by H2-H3 transition significantly narrows the Li-slab, making the Li-ion deintercalation considerably more difficult at higher SOC[26,27].

In contrast, cathodes with shorter sintering times—and consequently a higher RS phase fraction, which acts as a structural stabilizer—mitigate c-lattice shrinkage, thereby reducing polarization at higher SOC and enabling additional capacity above 4.3 V. However, as a trade-off, the cathodes with higher RS fractions display slightly increased polarization in the lower SOC range, due to the lower Li-ion conductivity of the RS phase, as shown in the charge/discharge curves (Fig. S12), leading to a modest sacrifice in overall rate capability. For comparison, we also prepared a single-phase LiNiO$_2$ sample by sintering the precursors at 700 °C for 6 h, hereafter referred to as s-LNO. Figure S13 demonstrates a similar rate dependence of specific capacity in the 600C-6h and s-LNO cathodes.

Subsequently, the 600C-6h sample was chosen as a representative dual-phase composite cathode to conduct more thorough electrochemical tests. For comparison, structural and morphological characterizations of the baseline s-LNO are shown in Fig. S14, with structural parameters summarized in Table S1. Additionally, its SEM images and first-cycle charge/discharge curves are presented in Figs. S9, S10, and S11, respectively, to allow direct comparison with the 600C-xh cathodes. As shown in Fig. 2c, the 600C-6h cathode delivers an initial capacity of 191.2 mAhg$^{-1}$ at a 2 C rate in an extended voltage range of 2.7–4.8 V vs. Li|Li$^+$, with an 88% retention after 1000 cycles—much higher than that of the s-LNO cathode (55.5% after 300 cycles). The corresponding specific energy, calculated based on the high average working voltage (3.86 V), is displayed on the secondary y-axis in Fig. 2c. It yields high energy densities of 738.0 Wh/kg in the first cycle and 603.4 Wh/kg in the 1000th cycle. Further insights about the improved structural stability of the 600C-6h cathode over the s-LNO are revealed by comparing the differential capacity (dQ/dV) curves of the two cathodes. As plotted in Fig. 2d, the dQ/dV profile of s-LNO shows multiple pairs of sharp anodic and cathodic peaks arising from the phase transitions among three hexagonal (H1, H2 and H3) structures and one monoclinic (M) structure. Sharp redox peaks in a dQ/dV plot represent the coexistence of two phases with distinct structures and thus abrupt structural changes during the phase transformation[6], which have been associated with the poor capacity retention of reported LiNiO$_2$ cathodes. In contrast, the dual-phase 600C-6h cathode displays distinctly broader and weaker peaks in the dQ/dV curve, especially above 3.7 V, indicating the suppression of structural changes in the active layered phase of the 600C-6h composite cathode. This is a key factor contributing to the improved cycling stability of the composite cathode at high voltages, consistent with low voltage polarization of charge/discharge curves during long-term cycling (Fig. 2e). Full-cell tests on 2 Ah pouch cells with the 600C-6h cathode were further performed against commercial graphite anodes, to evaluate the composite LNO cathode under more realistic industrial and pre-pilot-scale conditions. As shown in Fig. 2f, the 600C-6h||graphite full cell exhibits robust cycling performance with a capacity retention of above 90% at a voltage range of 2.7–4.2 V after 500 cycles and 80% in a further extended voltage window of 2.7–4.6 V after 1000 cycles at a high 1 C rate. The observed cycling stability of the 600C-6h cathode makes it a promising candidate for practical applications of high-performance cobalt-free LIBs.

In addition to the long-term cycling stability tests shown in Fig. 2c, f, we conducted electrochemical tests on the 600C-6h cathodes—using s-LNO as a baseline—under various charging rates and voltage ranges. For example, Fig. S15 presents tests in which both cathodes were cycled at a C/2 rate within voltage ranges of 2.7–4.4 V and 2.7–4.8 V. In all cases, the 600C-6h cathode demonstrated superior performance compared to s-LNO. The results are summarized in Table S2, alongside electrochemical test data from numerous LNO cathodes reported in the literature, including those modified by doping and/or coating processes. Notably, the dual-phase 600C-6h cathode, as a pure Li-Ni-O system, stands out for its robust performace among all listed cathodes.

## Fundamental mechanisms of the chemo-mechanical stability in dual-phase composite

To understand the effects of the interlock configuration of the 600C-6h composite on constraining lattice deformation during electrochemical cycling—and thereby preserving the structural integrity of secondary particles after extended cycling—we conducted *operando* SXRD and postmortem TXM examinations on both the dual-phase 600C-6h composite (d-LNO) and s-LNO cathodes. The two cathodes show distinct differences in their structural behavior that explain the observed contrast in their electrochemical performance. Figure 3a presents heatmaps of *operando* SXRD patterns from the d-LNO and s-LNO cathodes, recorded while charging from 3.4 to 4.8 V. The H2 to H3 phase transition region around 4.3 V is marked with dashed-line rectangles. While the diffraction peaks of the layered phase in both cathodes shift to higher angles in the H2-H3 region, the shifts in the s-LNO are much more pronounced (Fig. 3b, c). The abrupt peak position shifts in the s-LNO patterns near 4.3 V indicate a lattice collapse along the c-axis and a biphasic process due to the deep delithiation during the H2 to H3 transition[28,29]. In contrast, the peak shifts observed in the d-LNO are relatively moderate, with no visible splitting. For both cathodes, the (2–10) peak of the layered phase shifts only slightly, reflecting the insensitivity of the basal plane lattice to deep delithiation. This suggests that the $a–c$ lattice coupling at the coherent twin boundaries (Fig. 1f) may help stabilize the d-LNO in deeply charged state.

A quantitative account of the lattice changes resulting from the charging, derived from Rietveld refinements, is presented in Fig. 3d, e. In s-LNO, as the cell is charged from 4.08 V to 4.80 V, the lattice c shrinks from 14.41 Å to 13.41 Å, corresponding to an approximate 7%

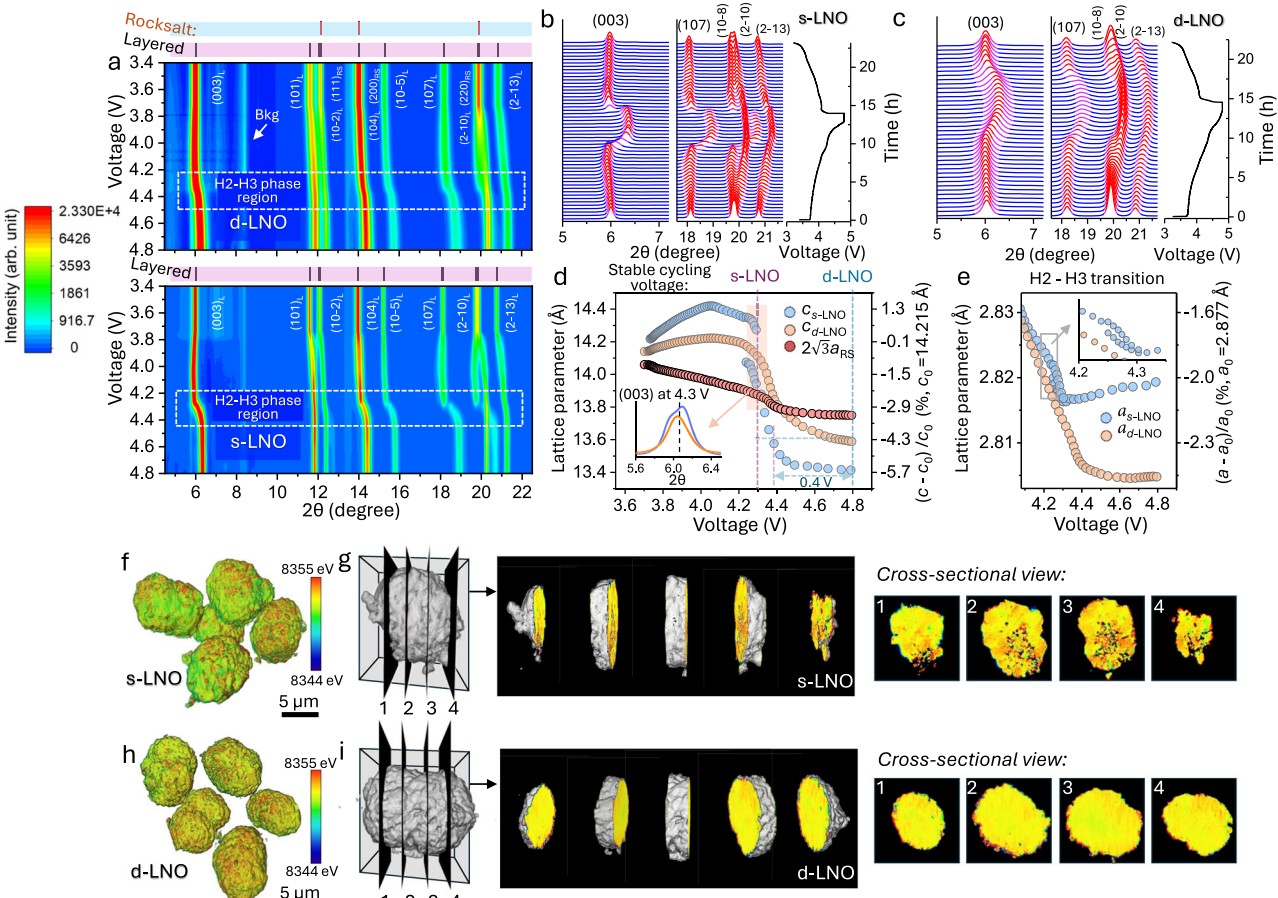

**Fig. 3 | Enhanced structural and mechanical stability of the dual-phase d-LNO (600C-6h) composite cathode compared to the single-phase s-LNO. a** Heatmaps of *operando* SXRD patterns recorded in the first cycle of the two cathodes at 0.1 C in a wide voltage range of 2.70–4.80 V vs. Li|Li⁺. The H2-H3 phase transition region is highlighted with dashed lines. For clarity, background signals from cell components other than the cathode were removed from the plots. **b, c** Magnified pattern showing the evolution of (003), (107), (018) and (2–10) reflections at high charging states during the first cycle of the two cathodes. **d** Variation of lattice-*c* of the layered phases in both cathodes, and lattice-*a* of the RS phase in d-LNO; the inset shows the 003 peak profiles at 4.3 V during the H2 to H3 phase transition. **e** Variation of lattice-*a* of the layered phases in both cathodes. The lattice parameters a₀ (2.877 Å) and c₀ (14.215 Å) of pristine s-LNO are used for calculating the lattice change percentage. TXM images showing three-dimensional reconstructions of cycled secondary particles for s-LNO (**f**) and d-LNO (**h**) after 200 cycles, together with internal structure features revealed by a series of cross-sectional slices on selected particles for s-LNO (**g**) and d-LNO (**i**), respectively. Source data for (**a, b, c, d, e**) are provided as a Source Data file.

reduction, while in d-LNO the lattice-*c* changes from 14.22 Å to 13.59 Å, over the same voltage range, corresponding to a smaller shrinkage of around 4 %. In contrast, the lattice-*a* in both d-LNO and s-LNO changes by less than 1% (Fig. 3e), indicating the anisotropic nature of the lattice distortion. Also shown in Fig. 3d is the variation of lattice a_RS of the rocksalt phase, scaled by a factor of 2√3 for comparison with the lattice *c* of the layered phases. The change in rocksalt lattice a_RS parallels the trend of the layered phase's lattice *c*, but to a much smaller extent, reflecting the elastic coupling between the rocksalt and layered phases. This coupling significantly suppresses severe lattice distortion, thereby maintaining lithium transport at deep charging states and alleviating the intrinsic kinetic limitations of high-Ni materials[30,31]. Additionally, the suppressed lattice-*c* contraction extends the stable operation of d-LNO to higher voltage. As shown in Fig. 3d, the lattice parameter *c* for d-LNO is 13.59 Å at 4.80 V, about 0.40 V higher than that of s-LNO for the same lattice shrinkage. Another important difference is that in s-LNO, an abrupt change in lattice-*c* from the H2 to H3 phase begins at 4.25 V, while in d-LNO, lattice-*c* changes continuously without a clearly defined phase transition voltage. The smoothed structural variation in d-LNO, as measured by SXRD, consistent with its dQ/dV profile (Fig. 2d). The substantial lattice mismatch in s-LNO between H2 and H3 phases, evidenced by the split of the 003 peak at

4.3 V (Fig. 3d, inset), lead to a high misfit dislocation density, generating fresh, reactive surfaces, and triggering lattice oxygen release and other irreversible reactions[32]. In contrast, the sudden lattice collapse is avoided in d-LNO due to elastic coupling between its rocksalt and layered domains (Fig. 1e), which suppresses the associated irreversible reactions and greatly improves structural stability during electrochemical cycles. Interestingly, the lattice parameter behavior in d-LNO is similar to that of reported high-Ni cathode with Co and Mn substitutions[33]. One plausible explanation for the suppression of H3 phase in NMC811 is that randomly distributed Co- and Mn-dopants inhibit long-range lithium/vacancy ordering and collective Jahn-Teller distortion. In d-LNO, devoid of any foreign element dopant, the same effects may be facilitated by the inactive rocksalt domains.

To further evaluate the effect of structural interlocking on the mechanical stability of dual-phase 600C-6h secondary particles, we conducted full-field TXM tomography on cathodes that had undergone long-term electrochemical cycling. Fig. 3f, h show the three-dimensional (3D) reconstructions of secondary particles of s-LNO and d-LNO after 200 cycles, respectively. Notably, an arbitrarily selected s-LNO secondary particle exhibits substantial intergranular nanocracks and nanopores within its internal structure, as shown in a series of cross-sectional slices (Fig. 3g), indicating severe chemo-mechanical

degradation after repeated lithiation/delithiation cycles. In contrast, no cracks are observed in the d-LNO particles (Fig. 3i). Cross-sectional SEM images (Fig. S11) reveal the presence of internal pores in pristine s-LNO and d-LNO, formed during the synthesis process. However, open nanocracks develop exclusively in the secondary particles of s-LNO. The slight differences in internal porosity formed during synthesis between the two cathodes are unlikely to be the primary factor driving nanocrack formation. Mechanical fracture in secondary particles primarily arises from the expansion or contraction of agglomerated primary particles in their original compact arrangement. The structural degradation of a primary particle results from anisotropic lattice variation caused by repeated extraction and insertion of lithium ions within the layered domains during electrochemical cycling[34–36]. By contrast, interlocking with the rocksalt component in the composite cathode suppresses the mechanical anisotropy of the layered component in the d-LNO, as evidenced by operando SXRD measurements (Fig. 3a–c). This suppression enhances lattice stability and structural integrity of the primary particle, thereby ensuring the morphological preservation of secondary particles.

The effect of rocksalt domains within the d-LNO cathode on its electrochemical performance is fundamentally different from that of the NiO-like rocksalt phase formed on the surface of NMC cathodes during electrochemical cycling, as reported in the literature[37,38]. The impact of the surface rocksalt phase on cathode performance is complex, often intertwined with the effects of the cathode/electrolyte interface layer, and generally degrades cycling stability, especially at high state of charge. In contrast, in our work, the rocksalt phase is intentionally formed during synthesis through kinetic control, resulting in a bulk-integrated dual-phase composite with an interlocked structure. This unique architecture suppresses lattice deformation during deep charging, thereby significantly enhancing the cycling stability of the d-LNO cathode.

To maintain high capacity retention during cycling, the d-LNO cathode must preserve its interlocked dual-phase crystal structure and morphology, as verified by post-cycle XRD measurements. Figure S16 and Table S4 present the SXRD patterns and refined structural parameters, respectively, for the d-LNO cathode before cycling and after 1 and 10 cycles. The RS fractions in these three samples range from 17 to 21%, showing no dependence on the number of cycles, likely due to the nonuniform RS distribution in the pristine powder. The LNO component retains its stoichiometric order, while the Li content in the RS phase and the LNO domain size remain consistent across all samples. This demonstrates that the dual-phase structure of the d-LNO cathode is preserved after multiple cycles and is a key factor contributing to its cycling stability.

The samples for post-cycle SXRD measurements were prepared by removing the cathode material, along with the aluminum current collector, from coin cells after a specific number of cycles. The cells were fabricated as described in the *Methods* section. SXRD patterns were collected with the cathode powder on the aluminum foil, following the procedure outlined in the *Methods* section. The 0-cycle sample was obtained from a cell that underwent no cycling. The 1-cycle and 10-cycle samples were pre-activated with two cycles at 0.1 C, followed by 1 and 10 cycles, respectively, at a C/3 rate within a voltage range of 2.7–4.8 V.

Additionally, HAADF-STEM imaging with atomic resolution was performed on the d-LNO cathode after 1000 cycles between 2.7 and 4.8 V at a 2 C rate (Fig. 2c). As shown in Figure S17, phase domains and boundaries are preserved even after prolonged cycling.

## Design generality of composite cathodes

The kinetically adjustable layered/rocksalt (L/RS) phase ratio in the dual-phase composite cathode provides a flexible approach for designing cathodes with desired capacity and cycling stability. As shown in the earlier section, at a constant temperature (600 °C), the

phase ratio for the best performance was identified using SXRD analysis of samples prepared with different heating times (Fig. 2b and Table S1). On the other hand, the selection of sintering temperature was based on the structure order of the $LiNiO_2$ phase, in addition to the phase ratio. Time- and temperature- resolved in-situ SXRD analysis as shown in Fig. 4a, b, shows that there are two forms of lithium nickelate[39]: one assumes a rocksalt structure $Li_xNi_{2-x}O_2$ with x less than or equal to 0.5, and the other a layered, close-to-stoichiometric $LiNiO_2$, coexisting within a temperature range from 475 to 640 °C. Above 640 °C, the layered nickelate becomes off-stoichiometric with observable nickel mixing at its 3b sites (Fig. 4c, upper panel). This in-situ temperature window for a highly ordered layered phase depends on the heating time due to the reaction kinetics[40]. A series of SXRD measurements were then conducted on samples heated at different temperatures for 6 h. The dual-phase fraction, Li occupancy and lattice parameters derived from these measurements are listed in Supplementary Information Table S4. 600 °C was chosen as it is the highest temperature at which the layered phase remains close-to-stoichiometric in samples measured after 6 h of heating. Maintaining close-to-stoichiometric $LiNiO_2$, as the electroactive component in the dual-phase composite cathode, is essential for achieving high reversible capacity, as $LiNiO_2$ with even moderate Ni mixing in the Li layers will deteriorate capacity retention[41]. Accompanying the high Li/Ni ordering in close-to-stoichiometric $LiNiO_2$ is its maximized Li-slab thickness, which occurs at about 600 °C, as shown in the lower panel in Fig. 4c. The greater Li-slab thickness favors high mobility of Li ions in the layered structure and is necessary for better electrochemical performance[26].

To further investigate the temperature effect on the kinetic control of the dual-phase composite, we conducted isothermal in-situ SXRD measurements on the system at 600 and 650 °C, as shown in Fig. 4d. As expected, higher temperatures accelerate the reaction kinetics. Both the layered phase fraction and domain size increase with heating time during the isothermal processes, as can be observed from the evolution of the $(003)_L$ peak intensity and width (Fig. 4d, side panels). The sample processed at the higher temperature (650 °C) outpaces that at the lower temperature (600 °C) in both respects (Fig. 4e). The layered phase fraction reaches 87% after 4 h at 650 °C and after 10.5 h at 600 °C. Clearly, the lower processing temperature offers greater tolerance in heating time for phase ratio adjustment, which is particularly advantageous for engineering control in industrial production. Interestingly, the average domain size in the sample processed at 650 °C—when the layered phase fraction reaches 87%—is 35 nm, larger than the 23 nm observed in the sample processed at 600 °C. The larger primary particle size of $LiNiO_2$, especially when accompanied by higher Ni/Li mixing, may lead to a reduction in capacity[42]. However, the size effects of $LiNiO_2$ in the $(LiNiO_2)_L|(Li_xNi_{2-x}O_2)_{RS}$ composite on electrochemical performance have yet to be fully explored. The isothermal in-situ SXRD analysis demonstrates that, to a certain extent, the domain size of $LiNiO_2$ can be controlled, along with the phase ratio—by adjusting the heating temperature and duration. Recent work on the NM9505 composite has shown that 5% Mn doping in the dual-phase composite significantly influences primary particle size control[9].

The rocksalt-to-layered conversion process forms a quasi-binary section, $k(LiNiO_2)_L|(1-k)(Li_xNi_{2-x}O_2)_{RS}$, in the ternary phase diagram of the Li-Ni-O system (Fig. S1). This reaction pathway is an unexpected finding, as previous studies identified the quasi-binary system as a single solid solution phase $Li_yNi_{2-y}O_2$, with $y = k + (1-k)x$, assuming equal Li/Ni ratios in both phases. The system converts to rhombohedral ($R\bar{3}m$) from a cubic ($Fm\bar{3}m$) structure at a composition boundary ($y \approx 0.62$) as expected in a thermodynamically equilibrated transition[15,17,43,44]. Our in-situ SXRD analysis of the sintering process reveals that the reaction does not follow the

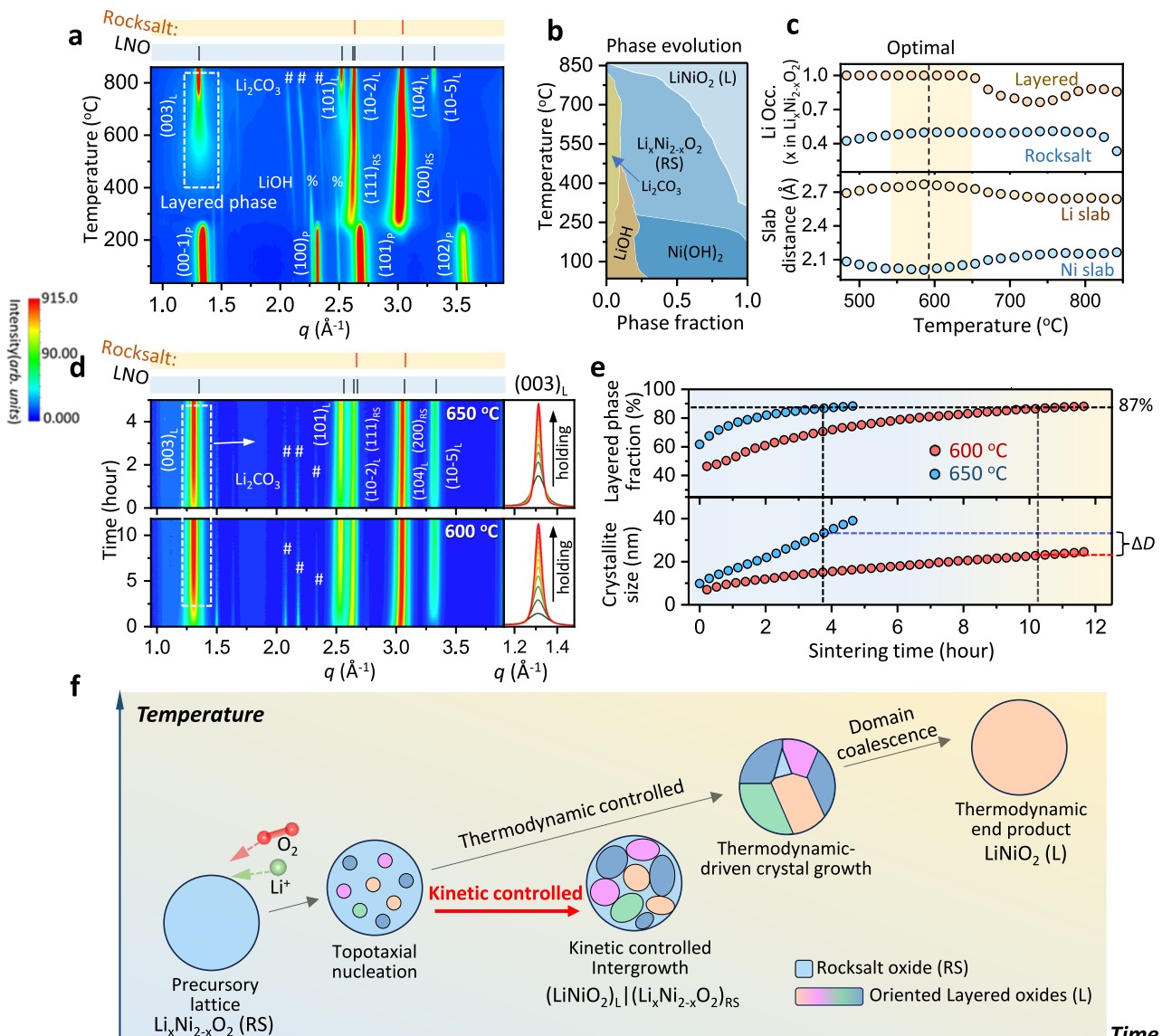

**Fig. 4 | Design approaches based on tuning the phase fraction, structure, stoichiometry, and domain size in the composite cathode via kinetic control.** **a** Heatmaps of SXRD patterns recorded during precursor sintering with linearly increasing temperatures. **b** Temperature-dependent phase fractions. **c** Lithium occupancy in the disordered rocksalt and layered phases, alongside the Li-/Ni-slab thickness of the layered phase as a function of temperature, derived from Rietveld refinement of the SXRD patterns. **d** Heatmaps of in-situ SXRD recorded under isothermal conditions at 650 °C (top) and 600 °C (bottom), depicting the propagation of the layered phase at two different sintering temperatures, with zoomed-in views of the peak evolution of the (003) reflection (side panels). **e** Layered phase fraction and crystal size variation during the two isothermal processes extracted from the SXRD patterns shown in (**d**) through quantitative refinement analysis. **f** Schematics illustrating the evolution of crystal phases in a primary particle of the Li-Ni-O system during the sintering process, highlighting kinetically and thermodynamically controlled reaction pathways. Source data for (**a**, **b**, **c**, **d**, **e**) are provided as a Source Data file.

lowest energy pathway. The $(LiNiO_2)_L|(Li_xNi_{2-x}O_2)_{RS}$ composite is a metastable intermediate, observed in the time-resolved in-situ measurements due to the slow reaction kinetics. The rocksalt-to-layered conversion in the equilibrium phase diagram was first described by Goodenough in 1958, using ex-situ XRD on samples annealed at 650 °C for 110 h[16]. In contrast, our SXRD measurements were conducted in situ, with temperature ramped from room temperature to 867 °C, spending only 55 min in the 500–650 °C window. It is generally observed that the intermediate phases formed during crystalline phase transition are not necessarily the most thermodynamically stable, as summarized by Ostwald's rule of stage[45]. While not a universal law nor rationalized by existing theory, the rule suggests that systems may pass through a series of metastable states before reaching equilibrium. Being able to explain and further predict such metastable reaction pathways

would represent major progress in the rational design of material synthesis, but remains an unmet challenge.

Quasi-binary sections in ternary or quaternary systems with adjustable phase ratios have been investigated in many materials synthesis studies[46–48]. For example, the quasi-binary system of (1-x) $Li(CB_9H_{10})|xLi(CB_{11}H_{12})$, used as a solid electrolyte, shows high electrochemical stability with $TiS_2$ cathodes, when $x = 0.3$[46]. However, the phase ratio in such systems is determined by the starting precursors, and the end products are thermodynamically stable, typically single-phase solid solutions. In contrast, the $LiNiO_2$ phase in our work is generated within the rocksalt phase, with a kinetically controllable phase ratio and crystal size, and the end product is kinetically stable —meaning it does not reach the lowest energy configuration at room temperature, but remain highly stable under practical application conditions.

The design principle of the dual-phase composite synthesis is schematically illustrated in Fig. 4f. The rocksalt NiO phase, formed during the decomposition of $Ni(OH)_2$ in the precursors, begins to be lithiated at its nucleation stage, and yields $Li_xNi_{2-x}O_2$[38]. When the x reaches 0.5, as observed in the in-situ SXRD measurement, a metastable dual-phase composite, $(LiNiO_2)_L|(Li_xNi_{2-x}O_2)_{RS}$, forms as an Ostwald-type intermediate. The L/RS phase ratio increases with temperature and heating duration, and eventually reaches a thermodynamically stable single solid solution phase $Li_xNi_{2-x}O_2$. The kinetically stable dual-phase composite, with an optimized phase ratio, is synthesized via kinetic control by rationally selecting the heating temperature and duration. The optimized phase ratio ensures high charge capacity and adequate concentration of interlocked domain boundaries to maintain structural stability. This synthesis strategy leverages kinetic control to produce kinetically stable materials through the rational selection of intermediate structures along metastable reaction pathways. By doing so, it significantly broadens the search space for novel functional materials. This approach—supported by in-situ SXRD structural analysis and TEM imaging—enables sophisticated manipulation of phase fraction, domain size, grain boundary, structure, and interfacial characteristics to optimize the material functionalities. It is applicable to a wide range of materials synthesis processes and is particularly well-suited for other high-Ni layered oxides with similar topotaxy-related dual-phase structures.

In conclusion, the interlocked $(LiNiO_2)_L|(Li_xNi_{2-x}O_2)_{RS}$ composite, with its phase ratio optimized through kinetic control, delivers high cycling stability with competitive capacity as a cobalt-free cathode for LIBs. The enhanced structural stability is attributed to the suppression of anisotropic lattice distortion in the active cathode material, as evidenced by *operando* SXRD and post-cycle TXM analysis. The interlock relation—manifested as coherent domain boundaries between rocksalt $Li_xNi_{2-x}O_2$ and layered $LiNiO_2$ domains, as shown in the scanning transmission electron microscopy (STEM) images—develops during the nucleation and growth of close-to-stoichiometric $LiNiO_2$ crystals within the rocksalt host lattice and originates from the topotaxial relationship between the two crystals. The dual-phase composite, evolving within a quasi-binary section of the Li-Ni-O ternary system as an Ostwald polymorph of the thermodynamically stable phase $(Li_yNi_{2-y}O_2)_L$, was identified through in-situ SXRD analysis and extracted via a kinetically controlled synthesis reaction. The synthesis strategy developed in this work—producing kinetically stable materials by monitoring and intercepting metastable reaction pathways—may find broad applications across various fields of materials synthesis.

## Methods

### Synthesis
The battery-grade $Ni(OH)_2$ precursor was provided by GEM Co., Ltd (Jingmen, China). Different Li-Ni-O samples were prepared by annealing the $Ni(OH)_2$ and LiOH (99.9%, Sigma Aldrich) mixture under various heat treatment conditions. To obtain the phase evolution diagram of LNO under thermodynamic conditions, 6 samples were prepared by calcining the mixture of $Ni(OH)_2$ and LiOH at a molar ratio of 1:1.05 at 400, 500, 600, 700, and 800 °C for 6 h in air, respectively. The dual-phase $(LiNiO_2)_L|(Li_xNi_{2-x}O_2)_{RS}$ composite materials with different rocksalt phase fractions were obtained by annealing the mixture of $Ni(OH)_2$ and LiOH at a molar ratio of 1:1.05 at 600 °C for 3, 6, 12, 18, 36, and 48 h under pure oxygen flow. For comparison, a single-phase $LiNiO_2$ sample was also synthesized by heating the same precursor mixture at 700 °C for 6 h under pure oxygen flow. The temperature ramps during heating and cooling in all the above calcination procedures were set to 2 °C/min.

### Electrochemistry
The electrode material was composed of 90 wt% cathode material, 5 wt% Super-P as the conductive carbon and 5 wt% polyvinylidene fluoride (PVDF) as the binder. The uniform slurry of the above three components was obtained using N-methyl-2-pyrrolidinone (NMP) as the solvent in a Thinky Mixer (ARE310) at 2000 rpm for 10 min under ambient air and room temperature. After casting the slurry onto aluminum foil (99.99%, thickness 12 μm; Yangzhou Nanopore Innovative Materials Technology Co., Ltd.) as the current collector using an MTI automatic coating machine (MSK-AFA-II-VC), the electrode was dried at 90 °C for 1 h in air, followed by 120 °C for 6 h under vacuum. The mass loading of active material in the cathode was ~5.5 mg/cm², with a 12 mm diameter cut using an MTI slicing machine (MSK-T10). The cathodes were assembled into two-electrode CR2032-type coin cells in an Ar-filled glovebox (MBraun, $O_2$ and $H_2O$ concentrations <5 ppm) for electrochemical measurements, with lithium foil (99.9%, 15.6 mm diameter, 0.45 mm thickness, Jiaxing Changgao New Material Technology Co., Ltd) as the reference and counter electrodes, and glass microfiber (Whatman, Grade GF/B, 675 μm thickness) as the separator. The LB062-type electrolyte was purchased from Duoduo Chemical Technology Co., Ltd., labeled as "4.6 V high-voltage electrolyte for lithium-ion batteries". Galvanostatic charge/discharge tests were performed on coin cells at various current densities in voltage ranges of 2.7–4.6 V and 2.7–4.8 V *vs.* Li|Li$^+$ using a BTS4000-5V battery testing system (Shenzhen Neware, China). The 1 $C$ rate corresponds to 180 m Ag$^{-1}$. Full pouch cells with capacity of 2 Ah were assembled using the dual-phase $(LiNiO_2)_L|(Li_xNi_{2-x}O_2)_{RS}$ as the active cathode material and the graphite as the anode material in a dry room with a dew point of −40 °C. The cathode consisted of the active cathode material, Super P, and PVDF in a weight ratio of 94:3:3, and the anode consisted of graphite, carboxymethylcellulose sodium, styrene butadiene rubber, and Super P in a weight ratio of 93.5:2:3:1.5. The loading of active $(LiNiO_2)_L|(Li_xNi_{2-x}O_2)_{RS}$ in the cathode was about 17.1 mg/cm², and the N/P ratio was set to 1.12. Galvanostatic charge/discharge tests were conducted on pouch cells at 1 $C$ (2 Ah) in voltage ranges of 2.7–4.2 V and 2.7–4.6 V using the same BTS4000-5V system. No extra pressure was applied on pouch cells during cycling in an environmental chamber maintained at 25 °C ± 0.5 °C under static heating.

*Operando* synchrotron XRD measurements during the first charge/discharge cycle of the two cathodes were carried out at NSLS-II beamline 7-BM (QAS) at Brookhaven National Laboratory (BNL). The $LiNiO_2$ and $LiNiO_2)_L|(Li_xNi_{2-x}O_2)_{RS}$ electrodes were prepared from a slurry of 90% active materials, 5% carbon black (Super P), and 5% PVDF (Kynar) in NMP and tape-cast onto carbon paper. These electrodes were fabricated into coin cells with Kapton X-ray window to avoid the X-ray absorption and scattering from the coin cell case. The electrolyte was 1 M $LiPF_6$ in EC/EMC/DEC (1:1:1 by volume) with <10 ppm moisture. Cells were cycled using a Biologic VSP multi-channel testing system from 3.0 to 4.8 V at a C/10 rate (20 m Ag$^{-1}$). The X-ray diffraction pattern of NIST $CeO_2$ standard was measured under identical conditions for instrumental calibration. Diffraction patterns were collected every 10 min using a PerkinElmer area detector during the first charge/discharge cycle of each electrode. The wavelength of the X-ray beam was 0.49594 Å. The focused spot size was 1.5 mm (vertical) × 0.5 mm (horizontal).

### In-situ and ex-situ synchrotron XRD measurements
Time-resolved in-situ synchrotron XRD patterns were collected at beamline 28-ID-1of the National Synchrotron Light Source II (NSLS-II) at BNL. The wavelength of the X-ray beam was 0.1666 Å, and the beam spot size at the sample was 0.5 mm (horizontal) × 0.5 mm (vertical). A powder pellet composed of $Ni(OH)_2$ and LiOH at a molar ratio of 1:1.05 was placed in a vertically mounted Linkam TS 1500 furnace, with an open window perpendicular to the X-ray beam. The pellet was heated in 20 °C steps from 35 °C to 1000 °C (nominal control temperature), held for ~5 min at each step, and measured using a 30-s exposure per

XRD pattern. An amorphous silicon flat-panel X-ray detector (PerkinElmer XRD 1621) was used to collect the XRD patterns. A temperature calibration curve correlating sample temperature to the nominal control temperature was obtained by measuring the thermal expansion of $CeO_2$ under identical conditions. Ex-situ synchrotron XRD patterns were recorded using powder samples filled in Kapton capillaries, collected at beamlines 28-ID-1 and 28-ID-2, with X-ray beam wavelengths specified in the main text.

### Synchrotron TXM measurements
Synchrotron-based transmission X-ray nanotomography was performed at beamline 18-ID (FXI) of NSLS-II, BNL. Individual projection images were captured with an exposure time of 0.05 s. Tomographic data were collected in fly-scan mode with a fixed exposure time of 0.05 s, while the sample was rotated continuously from 0° to 180° at a typical speed of 3°/sec.

### Refinement of SXRD data
Rietveld refinements of SXRD patterns were performed using TOPAS-A V5 software. Global refinement parameters included background coefficients, peak shape parameters, lattice parameters, and the weights of different structural phases. The $U_{iso}$ values of elements occupying equivalent crystallographic positions were constrained to be equal.

### Electron microscopy characterization
Morphology and structure of the samples were examined by scanning electron microscopy (SEM, Hitachi S-4800), operated at 20 kV, and transmission electron microscopy (TEM, FEI Tecnai G2T20), operated at 200 kV. Elemental mappings were recorded using energy-dispersive X-ray spectroscopy in STEM mode.

## Data availability
Source data are provided with this paper.

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

## Acknowledgements

This work was supported by the General Program of Natural Science Foundation of Jiangsu Province in China with the grant no. BK20221238. This research used 7-BM (QAS), 18-ID (FXI), 28-ID-1 (PDF), 28-ID-2 (XPD) beamline at the National Synchrotron Light Source II, a U.S. DOE Office of Science User Facility operated for the DOE Office of Science by Brookhaven National Laboratory under contract no. DE-SC0012704. J.Z. and J.Y. were supported by the National Natural Science Foundation of China with the grant no. 21703147. L.C. was supported by the National Key R&D Program of China with the grant no. 2021YFB3800300. S.L. and F.W. were supported by the U.S. Department of Energy (DOE) Office of Energy Efficiency and Renewable Energy, Vehicle Technologies Office, contract no. DE-SC0012704. L.W. was supported by the U.S. Department of Energy, Office of Basic Energy Science, Division of Materials Science and Engineering, under Contract No. DE-SC0012704.

## Author contributions

J.Z., F.W., and J.B. conceived the idea and designed the work; J.Y., S.L., W.Z., Z.Z., P.H., M.G., L.M., X.Z., K.X., and K.Z. carried out the sample preparations, material characterizations and electrochemical performance tests. W.Z., Y.S., L.C., L.W., S.L. and J.Y. performed the STEM measurements and analysis; S.L., J.B., and F.W. performed the temperature-dependent synchrotron XRD measurements and analysis; S.L. and L.M. performed the isothermal synchrotron XRD measurements and analysis; S.L. and M.G. performed the synchrotron TXM measurements and analysis; J.Y., S.L., and J.B. performed the Rietveld Refinement analysis; S.L., J.Y., J.Z., and J.B. performed *operando* synchrotron XRD measurements during electrochemical cycles and analysis; Z.Z., H.P., and J.Z. assembled the full cell and carried out electrochemical tests and analysis. K.X. and K.Z. prepared the precursor. J.Y., S.L., J.Z., F.W., and J.B. wrote the manuscript with critical input from all other authors. All authors edited and approved the manuscript.

## Competing interests

The authors declare no competing interests.
