## [Transparent Peer Review file · Nature Communications]

Topotaxially Grown Composite Cathodes for Cobalt-free High-Energy Long-Life Li-ion Batteries

Corresponding Author: Dr Jianming Bai

Version 0:

Reviewer comments:

Reviewer #1

(Remarks to the Author)

This manuscript reports on a two-phase composite cathode with enhanced performance for lithium-ion batteries. The authors claim that the cathode consists of a layered R-3m phase and a rocksalt phase in an 87:13 weight ratio. However, based on the XRD and in-situ SXRD results, the reflections of the rocksalt phase appear to be completely overlapped with those of the parent R-3m phase. While 13 wt.% of rocksalt is indeed significant, the observed Li/Ni mixing raises concerns, together with the serious overlapping in XRD, making the authors' claim less convincing. In addition, the novelty of this work is insufficiently presented. Although the performance is promising, considering the standards expected for Nature Communications, I cannot recommend acceptance of this work.

Reviewer #2

(Remarks to the Author)

This manuscript presented a strategy to enhance the structural integrity of LiNiO₂ (LNO) by optimizing an interlocked dual-phase composite to address the cycling instability issue. The authors emphasized that regulating the elastic coupling between the layered and rock-salt phases through kinetic control during sintering could mitigate the anisotropic lattice distortion during high-voltage electrochemical cycling, resulting in improved cycling stability by alleviating structural change. The reviewer agrees with the significance of this research in mitigating the degradation of LNO cyclability by synthesizing a heterostructure based on the principles of synthesis rather than relying on conventional doping approaches. However, this manuscript lacks sufficient evidence to support its claims and would benefit from the inclusion of additional data to strengthen the arguments before being considered for publication in Nature Communications.

1. The structural analysis of the pristine 600C-6h LNO sample was presented using SXRD and TEM data in Figure 1. However, these two methods alone did not provide a clear view of whether the rock-salt phase was distributed on the surface or within the bulk. Therefore, I recommend using TXM-XANES imaging to determine the oxidation state of Ni ions and to visualize the distribution of the rock-salt phase (Nat Commun 15, 430 (2024)).
2. In Figure 2b, the authors reported that Rietveld refinement results showed a decrease in the rock-salt phase fraction with increasing sintering time, and they observed a trend where higher rock-salt fractions led to decreased initial capacity but improved cycling stability. They argued that an increase in the rock-salt phase fraction resulted in more coherent boundaries, thereby enhancing structural stability. However, factors such as particle size, crystallite size, and morphology, which could also change with sintering temperature and time, may have influenced the observed cycling stability. In fact, the Rietveld refinement results in Table 1 indicated changes in crystallite size, which could also contribute to the improved cycling stability. Therefore, I recommend that the authors provide an explanation and additional data on these excluded factors (e.g., cross-sectional SEM images to verify particle size and morphology) (ACS Appl. Mater. Interfaces 2024, 16, 23150–23159).
3. To evaluate the effectiveness of the interlock configuration in the 600c-6h composite, the authors synthesized s-LNO as a control. Although Rietveld refinement results were provided in Table 1, there was insufficient data to confirm whether this control was reasonable. First, please provide HAADF-STEM images and SAED patterns to confirm that the synthesized s-

LNO was indeed a single phase. Second, please provide SEM images showing particle morphology and size. Lastly, include the raw data from the Rietveld refinement.

4. The SXRD data in Figure 3b showed that the phase transition during the first charge and discharge was asymmetric. The phase transition occurred at high voltage during charging, while it occurred at relatively low voltage during discharging. Additionally, the voltage profiles for the first cycle were missing in Figure 2e. This suggested that phase transitions in the first cycle might differ from those in subsequent cycles. This reviewer requests that the authors provide the voltage profiles for the first cycle of the 600oC-6h sample. If the phase transition occurred differently in the first cycle compared to subsequent cycles, please illuminate the reason for this.

5. Figure 2f presented cycle test results for full cells assembled as pouch cells, tested with different cut-off voltages to assess stability under conditions similar to commercial use. However, the data was presented in terms of capacity retention, without providing the actual capacity. Different cut-off voltages would likely result in varying capacities, and it is necessary to show these differences in terms of actual capacity. Additionally, for the sample tested within the 2.7 V–4.6 V voltage range, the initial capacity retention exceeded 100%, indicating some form of activation or another cause in the early cycles. Please illuminate the reason for this observation.

6. Figure 2a showed cycle test results for samples synthesized with different sintering times at 600°C. While this data presented capacity changes over cycles, it did not show how the voltage profiles differed depending on sintering time. Please include information on how phase changes varied with sintering time, including corresponding voltage profile data.

7. The manuscript claimed that interlocking with the rock-salt component in the cathode composite alleviated the mechanical anisotropy of the layered component after electrochemical cycling, which was supported by SXRD data. However, please provide TEM images with atomic resolution to directly and intuitively observe any changes at the boundary between the rock-salt and layered phases after cycling.

8. In Figures 3f-i, the formation of intergranular nanocracks in s-LNO and d-LNO was identified using 3D TXM imaging. However, differences in the size or distribution of pores between primary particles in pristine LNO could be a cause of this observation. To confirm that nanocrack formation in s-LNO and d-LNO was due to the elastic coupling between the layered and rock-salt phases, please demonstrate that the secondary particles in pristine s-LNO and d-LNO had similar morphology and porosity using 3D TXM images before cycling.

9. Please add a section on Transmission X-ray Microscopy (TXM) in the Methods section.

10. The Rock-salt fraction (%) data for the 600oC-3h sample in Figure 2b did not match the Weight Fraction data for the rock-salt phase in Table 1. Please correct this discrepancy.

Reviewer #3

(Remarks to the Author)

I appreciate the author's efforts in investigating the electrochemical performance of the mixed layered and rock-salt LiNiO₂ composite cathode. Showcasing a cobalt-free cathode with high energy and long cycle life. However, there are serious concerns regarding the thorough understanding of this intermediate phase. This study also follows up on the layered LiNi_{0.95}Mn_{0.05}O₂ and rock salt intermixed phases (<https://doi.org/10.1038/s41467-023-44583-3>), which raises questions about the novelty of this study. The title of the paper is attractive, but I didn't find it relevant. The authors didn't control any growth orientation of the particles or preserve original structure of LNO.

1. Why authors are claiming it as a new nickelate cathode? The Pseudo phase diagrams (of Li₂O-O-NiO) in earlier studies mentioned the intermediate phases. Please refer to Matteo Bianchini et al., (<https://onlinelibrary.wiley.com/doi/10.1002/anie.201812472>).

2. Is the same phase reproducible at that particular temperature and time with the same ratio of 13: 87 of Rs-LNO and L-LNO? How many times did the authors test its performance, repeating from the synthesis to cycling performance? The growth kinetics of these intermediate phases are influenced by many factors; sometimes, they are unstable under ambient conditions and tend to form a stable phase or decompose. The growth orientation of these metastable intermediate phases can also depend on the crucibles used. Did the authors face any issues while handling dual-mix cathode?

3. Does this mixed phase have any impact on the diffusion kinetics of the Li-ion migration? The authors didn't talk about kinetics.

4. This study is missing a performance comparison with reported literature. It seems the authors control S-LiNiO₂ is not up to standards. Compare with the best LNO perform from the literature.

5. The authors claim that interlocking LiNiO₂ with rack salt suppress the severity of the H₂ to H₃ phase transition. However, there is still, a considerable change in lattice parameter *c* for d-LNO = 0.8 and s-LNO = 1 Å. Surprisingly, TXM doesn't show any interparticle cracking for Rs-LNO. The authors should elaborate on their explanation. How can interlocking change the isotropic or anisotropic expansion of the layered portion of the particles (which is 87%)?

6. Why are the authors experiencing a huge IR drop in the cycling, especially during the discharge it is around 500 mV, as shown in Figure 2e?

7. Did the authors use O₂ gas for the synthesis of LNO? Generally, oxides are synthesized under oxygen flow to obtain good quality cathodes.

8. Space group symbols are not written properly; the letter should be in italics.

9. The authors mentioned that "even moderate Ni mixing in the Li-Layer will deteriorate capacity retention". The reported

material contains 87% of layered structure with small portion of rock-salt. How is long-term cycling stability protected in this material? How does this rock salt structure protects anti-site mixing in the layered structure over the 1000 cycles? There is reason if dopants helping in stabilizing the structure, but I couldn't see relevance with interlocking.

10. The authors mentioned that LNO started nucleating in the rock salt phase with different orientations. Can authors elaborate on this? What is the nucleation plan?

Reviewer #4

(Remarks to the Author)

The manuscript from Yao and Bai et al. presented a new explanation of the reaction mechanism of LiNiO₂ synthesized at low temperatures and at the short period of holding time. In the manuscript, the authors suggest a composite structure instead of a solid solution is formed at a non-equilibrium synthesis state. And the author tried to use some state-of-the-art analytics tools to resolve the mystery behind this. I believe this work is meaningful from academic perspective, but not from an industry perspective, given the lack of proper comparison due to lacking of care design of some control experiments, I would consider this manuscript potentially valuable but the authors need to make major revisions on many points to make this work meet the standard of Nature Communications. I recommend the authors to consider the following points:

1. It is difficult to distinguish if the material synthesized at 600°C-6h is a single phase or composite structure, one of the possible indicators is to measure and perform an in-depth analysis on the XRD pattern. Particularly the authors should focus on the "asymmetry" of the XRD peaks, please include this in your analysis.
2. To follow up on question 1, for the Rietveld refinement in Fig. 1a, how was the reference of the rock-salt phase Li_xNi_{2-x}O₂ chosen, what is the x value, because it will impact the lattice parameters and therefore the ratio of this rock-salt phase.
3. Figures 1c-g show the HAADF-STEM and FFTs images of this composite material, however, one of the drawbacks of using this kind of high-resolution microscopy analytics method is it can only show the composition of the selected of interest within a very small area, as indicated by the scale bar of these images. My concerns regarding these images are, first, how was the images chosen to display. Second, I believe that for LiNiO₂ with even the least amount of cation mixing, you would still observe rock-salt layers, I agree the mass ratio of the rocksalt phase should be way less than 13%, but do they perform any difference in terms of the mechanism of providing structural and electrochemical stability? Please perform a similar analysis and add an in-depth discussion on "normal" LiNiO₂ as a comparison.
4. The Voltage–capacity curves of LNO-600°C 3-48h can be added to supplementary documents to compare the difference and reason behind this.
5. The cycling stability of LiNiO₂ is heavily determined by the amount of Li (x in Li_{1-x}NiO₂) being utilized, please add long-term cycling comparison with LiNiO₂ at 700°C-6h that only utilizes the capacity (low voltage) to the same level as composite cathode made at 600°C-6h (~195 mAh/g). Do the authors still expect a similar conclusion that this composite structure material will still outperform its competitor?
6. In Figure 2c,d, it is not fair to test s-LNO with an upper cut-off voltage of 4.8V, there is virtually no capacity above 4.3V for s-LNO and it only accelerates the parasitic reaction between s-LNO and electrolyte.

Version 1:

Reviewer comments:

Reviewer #1

(Remarks to the Author)

I have carefully reviewed the revised manuscript and the authors' responses to the comments from all four reviewers. Overall, the authors have addressed the feedback in a reasonably comprehensive manner. However, as highlighted in Reviewer #3's Comment #2, the reproducibility of the results under the specified temperature, duration, and Rs-LNO to L-LNO ratio of 13:87 remains a concern. While the authors provide a definitive verbal assertion, such certainty is impractical given the complexity of sintering parameters. To substantiate this claim, the authors should present supporting evidence. Although the revised manuscript is generally reasonable, considering the high standards of *Nature Communications*, I am hesitant to recommend acceptance at this stage.

Reviewer #2

(Remarks to the Author)

The authors have addressed all of my concerns. The methodology used to analyze their data is now much clearer. I believe the manuscript should be accepted for publication.

Reviewer #3

(Remarks to the Author)

I sincerely appreciate the authors' efforts in addressing my earlier comments and acknowledge the significant improvements in the manuscript. I believe the study presents valuable insights and merits consideration for publication.

However, I have reservations regarding the response to my earlier Comment 3, which addressed the influence of the rock-salt structure on diffusion kinetics. This aspect was not sufficiently addressed.

The authors convincingly present the idea of interlocking and controlling the growth of the layered structure from the rock-salt phase through a kinetically controlled process. The advanced characterization techniques used to demonstrate the phase composition—87% layered and 13% rock-salt.

Nonetheless, previous studies, including a highly influential work published by Prof. Clare P. Grey's group in *Nature Materials* (Xu, C., Märker, K., Lee, J. et al., 2021; <https://doi.org/10.1038/s41563-020-0767-8>), suggest that the rock-salt phase can restrict Li-ion mobility in Ni-rich layered cathodes. The findings in this manuscript appear contradictory to that established understanding.

It is crucial for the authors to provide a more detailed and comprehensive discussion addressing this apparent contradiction, including potential differences in experimental conditions or mechanistic interpretations. This will enhance the scientific rigor and clarity of the manuscript.

Reviewer #4

(Remarks to the Author)

All my concerns and suggestions were properly answered, therefore the work meets my requirement for being accepted by the Journal.

Response to the Reviewers:

We sincerely appreciate the reviewers' time and effort in evaluating our manuscript. We have carefully considered their comments and given thoughtful attention to their feedback. In response, we have provided detailed, itemized replies to each of their questions, comments, and suggestions. Additionally, we have made the necessary revisions to the manuscript to address their concerns and further enhance the quality of the paper. All changes in the revised manuscript copied to this response are highlighted in red for clarity.

Our point-by-point responses to the reviewers' feedback are detailed below

Reviewer #1 (Remarks to the Author):

This manuscript reports on a two-phase composite cathode with enhanced performance for lithium-ion batteries. The authors claim that the cathode consists of a layered $R\bar{3}m$ phase and a rocksalt phase in an 87:13 weight ratio. However, based on the XRD and *in-situ* SXRD results, the reflections of the rocksalt phase appear to be completely overlapped with those of the parent $R\bar{3}m$ phase. While 13 wt.% of rocksalt is indeed significant, the observed Li/Ni mixing raises concerns, together with the serious overlapping in XRD, making the authors' claim less convincing. In addition, the novelty of this work is insufficiently presented. Although the performance is promising, considering the standards expected for Nature Communications, I cannot recommend acceptance of this work.

Response:

We thank the reviewer for taking the time to review our manuscript and provide valuable comments. In the following response, we address all the questions raised by the reviewer and have added more details to the revised manuscript, particularly regarding the synchrotron X-ray diffraction (SXRD) analysis of the dual-phase structure, to address the reviewer's concerns about the peak overlapping. Additionally, we have highlighted the key novelties of our work, as the reviewer suggested these were not sufficiently presented.

First, the layered $R\bar{3}m$ phase is not the parent phase of the rocksalt phase. On the contrary, both *in-situ* SXRD and HAADF-STEM results clearly show that the layered phase forms within the rocksalt phase, which itself emerges during the decomposition of hydroxides in the earlier stage solid-state reaction. This relationship was depicted in Figure 4f and verified by *in-situ* SXRD measurement during precursors sintering, together with the refinement analysis (Figure 4a-c). Second, the presence of 13 wt.% rocksalt does not imply an observable Li/Ni mixing in the layered phase. Based on the dual-phase model, Rietveld refinement reveals a close-to-stoichiometric layered phase (Figure 4c). This highly ordered layered structure, as the electroactive component in the dual-phase composite cathode, is primary contributor to its high energy density. As emphasized throughout the manuscript, the lithium-containing rocksalt phase stabilizes the electroactive layered phase during electrochemical cycling through coherent boundaries between the two crystalline components. This

was demonstrated by the HAADF-STEM images (Figure 1c-g) and the *operando* SXR measurements during the electrochemical charging (Figure 3a-e), which revealed a stark contrast in the behavior of the dual-phase d-LNO composite compared to the baseline single phase s-LNO with a Li/Ni mixing of about 0.5% (Table S1).

The overlap between the Bragg peaks of the rocksalt and layered phases does indeed make it challenging to identify the dual-phase composite. However, by carefully analyzing the peak profiles and relative intensities, and tracking the phase evolution trend in *in-situ* studies, we were able to unequivocally determine the dual-phase structure. The ability to correctly identify and further optimize this dual-phase structure with the aid of quantitative analysis of *in-situ* SXR data highlights one of the novelties in this work. In the previous version of the paper, we did not provide an extensive discussion of the SXR analysis. Instead, we referred to the derivation of the dual-phase model in our preprint available on Research Square (Ref. 30), as the cross-sectional HAADF-STEM image in Figure 1c already provided strong support for the dual-phase model. In response to the reviewer's comments, we have now added a section titled "Notes on the SXR Data Analysis" in the Supplementary Information (SI) to elaborate on the details of the SXR structural analysis.

As shown in Figures 4a and 4b, the rocksalt phase, *i.e.*, $(\text{Li}_x\text{Ni}_{2-x}\text{O}_2)_{\text{RS}}$ formed first upon decomposition of the hydroxide precursor, followed by the emergence of the layered phase, *i.e.*, $(\text{LiNiO}_2)_{\text{L}}$ at around 400°C , as indicated by the appearance of the 003_{L} peak. At this stage, the layered phase exhibited weaker and broader peaks compared to the rocksalt phase, making the dual-phase model the most suitable representation of the material (refer to Figure S4 in the revised SI and its corresponding discussions, and included here as Figure R1).

Fig. R1 | Three structural models used in the Rietveld refinement with respect to the SXR pattern recorded at 483°C during *in-situ* measurements. a, the layered structure with a $R\bar{3}m$ space group. b, the layered structure with a $R\bar{3}m$ space group and hkl dependent anisotropic peak

broadening. The broad (003) peak fits better but the fitting for higher angle gets worse. c, the dual-phase model composing of the layered phase with a $R\bar{3}m$ space group and the Li-containing disordered rocksalt phase with a $Fm\bar{3}m$ space group, revealing the best fitting result compared to the other two models presented in a and b. In the plots, blue circles are used for the observed data, red lines for the calculated data, colored (green, navy and orange) bars for the Bragg peak positions, and black lines for the difference between the observed and calculated patterns.

The coexistence of rocksalt and layered phases at these temperatures has been previously recognized, as reported in *J. Mater. Chem. A*, 2020, 8, 1808-1820 (Ref. 17 in the revised manuscript). However, no reliable quantitative analysis of the phase ratio was provided in that work, as it assumed a conventional polymorphic phase transition model, where the rocksalt and layered phases were presumed to have similar lithium content.

The key novelty of this work is the unexpected discovery of a miscibility gap in the Li-Ni-O system, revealing two separated phases with different structures and lithium occupancies. Using the composition formula $\text{Li}_x\text{Ni}_{2-x}\text{O}_2$ to represent the lithium nickelate compound, where the two ends of the solid solution are NiO rocksalt ($x = 0$) and the stoichiometric layered phase LiNiO_2 ($x = 1$), we identified this miscibility gap through quantitative analysis of *in-situ* SXR data, further corroborated by HAADF-STEM imaging. The gap spans a wide temperature range (475 to 640 °C), with $x \sim 0.5$ in the rocksalt phase and $x \sim 1$ in the layered phase (Figure 4a-c). This contrasts with the conventional view that the rocksalt-to-layered transition follows a continuous solid solution route with a polymorphic transition at $x = 0.62$ (*Phys. Rev. B*, 1992, 46, 3236. Ref. 15 in the revised manuscript). To the best of our knowledge, such a miscibility gap has not been previously reported for the Li-Ni-O system, even in the most recent *in-situ* XRD studies (*J. Mater. Chem. A*, 2020, 8, 1808-1820. Ref. 17 in the revised manuscript).

Figure R1 shows the *in-situ* SXR data analysis at 483 °C, the initial stage of the miscibility gap. At higher temperatures and with extended sintering times, the layered phase became the dominant component of the composite. Nonetheless, the dual-phase model remained valid for describing the Li-Ni-O system. This is supported by X-ray crystallographic data, detailed in the "Notes on the SXR Data Analysis" section added to the revised manuscript's SI, and by HAADF-STEM images (Figures 1c and S1).

This finding has two significant implications for Li-ion battery applications. First, the dual-phase composite—with the inactive rocksalt phase acting as a structural stabilizer and the near-stoichiometric layered phase serving as the electroactive component—enables extended cycle life when used as a cathode material. Second, due to the broad temperature range, the mass ratio between the two phases can be kinetically adjusted to optimize electrochemical performance (Figures 2a and 2b). The high capacity retention over long-term cycling, as demonstrated in Figures 2c and 2f, highlights both the importance and the novelty of this work.

The corresponding description has been added in the revised manuscript and revised Supplementary Information.

In addition to the "Notes on the SXR D Data Analysis" section in the Supplementary Information to elaborate on the details of the SXR D structural analysis, we included the following discussion in the "The interlocked rocksalt-layered composite originated from topotaxial crystal growth" section, to emphasize the novel discovery of the miscibility gap and its implications.

The crystallographic analysis of the dual-phase structure presented some interesting complexities, as explained in the 'Notes on the SXR D Data Analysis,' in the Supplementary Information, with details on the Rietveld refinement-based structural and quantitative analysis during the sintering process of the Li-Ni-O system.

.....

The dual-phase composite exhibits a miscibility gap in the Li-Ni-O system, characterized by two distinct phases with different structures and lithium occupancies. This finding stands in stark contrast to the conventional polymorphic phase transition model, which assumes similar lithium content in the rocksalt and layered phases. This insight was instrumental in enabling the synthesis of the dual-phase composite, with tuned components and structural properties for use as a high-performance cathode material in Li-ion batteries.

Reviewer #2 (Remarks to the Author):

This manuscript presented a strategy to enhance the structural integrity of LiNiO₂ (LNO) by optimizing an interlocked dual-phase composite to address the cycling instability issue. The authors emphasized that regulating the elastic coupling between the layered and rock-salt phases through kinetic control during sintering could mitigate the anisotropic lattice distortion during high-voltage electrochemical cycling, resulting in improved cycling stability by alleviating structural change. The reviewer agrees with the significance of this research in mitigating the degradation of LNO cyclability by synthesizing a heterostructure based on the principles of synthesis rather than relying on conventional doping approaches. However, this manuscript lacks sufficient evidence to support its claims and would benefit from the inclusion of additional data to strengthen the arguments before being considered for publication in Nature Communications.

1. The structural analysis of the pristine 600C-6h LNO sample was presented using SXR D and TEM data in Figure 1. However, these two methods alone did not provide a clear view of whether the rock-salt phase was distributed on the surface or within the bulk. Therefore, I recommend using TXM-XANES imaging to determine the oxidation state of Ni ions and to visualize the distribution of the rock-salt phase (Nat Commun 15, 430 (2024)).

Response:

We sincerely thank the reviewer for their insightful comments.

The SXR D pattern of the 600C-6h sample, as shown in Figure 1a, along with the refinement analysis, confirms the coexistence of rocksalt and layered phases in the (LiNiO₂)_L(Li_xNi_{2-x}O₂)_{RS} composite. We would like to kindly clarify that the HAADF-STEM image in Figure 1c was captured using a FIB (Focused Ion Beam)-thinned specimen, to show the dual-phase configuration inside the primary particle.

To provide further clarity, we include a HAADF-STEM image of a sliced primary particle (Figure R2a) and the corresponding high-resolution image from a selected region (highlighted by the yellow square) inside the particle (Figure R2b). Notably, in Figure R2b, the white dashed line indicates the area where the image in Figure 1c was captured, clearly demonstrating that the rocksalt phase is distributed within the bulk structure of the primary particle.

Furthermore, *in-situ* SXR analysis (Figure 4a-b) indicates that the rocksalt phase forms during the decomposition of the hydroxide precursor and is followed by the formation of the layered phase at higher temperatures, as lithiation occurs from an external lithium source. Thus, it is unlikely that the rocksalt phase is distributed on the particle surface, as lithium ions diffuse into the rocksalt particles from outside, especially at lower temperatures where the rocksalt phase dominates the composite. This mechanism differs from the formation of the rocksalt phase on the surface of the layered $(\text{LiNiO}_2)_L$ at high sintering temperatures, where lithium and oxygen loss occurs at grain boundaries.

We agree with the reviewer's suggestion that TXM-XANES imaging could be a powerful tool for determining the oxidation state of Ni ions and visualizing the rocksalt phase distribution. However, the combination of SXR and the sectional HAADF-STEM results provided here sufficiently verifies the distribution of the rocksalt phase within the bulk structure of the $(\text{LiNiO}_2)_L|(\text{Li}_x\text{Ni}_{2-x}\text{O}_2)_{RS}$ composite.

Fig. R2 | HAADF-STEM images of FIB-thinned $(\text{LiNiO}_2)_L|(\text{Li}_x\text{Ni}_{2-x}\text{O}_2)_{RS}$ composite obtained by sintering at 600 °C for 6 h. a, STEM image of a sliced primary particle prepared by the FIB. b, high-resolution HAADF-STEM image captured in a selected region of the primary particle as marked by the white arrow and yellow square in a. The lined area in b is selected to show the typical interlocked rocksalt and layered phases in such as a composite material as presented in Figure 1c.

2. In Figure 2b, the authors reported that Rietveld refinement results showed a decrease in the rock-salt phase fraction with increasing sintering time, and they observed a trend where higher rock-salt fractions led to decreased initial capacity but improved cycling stability. They argued that an increase in the rock-salt phase fraction resulted in more coherent boundaries, thereby enhancing structural stability. However, factors such as particle size, crystallite size, and morphology, which

could also change with sintering temperature and time, may have influenced the observed cycling stability. In fact, the Rietveld refinement results in Table 1 indicated changes in crystallite size, which could also contribute to the improved cycling stability. Therefore, I recommend that the authors provide an explanation and additional data on these excluded factors (*e.g.*, cross-sectional SEM images to verify particle size and morphology) (ACS Appl. Mater. Interfaces, 2024, 16, 23150-23159).

Response:

We appreciate the reviewer's insightful comments. The primary aim of Figure 2b is to illustrate that the phase ratio can be kinetically controlled and to explore its relationship with initial capacity and capacity retention. We attribute this effect to coherent boundaries, as supported by *operando* SXR, which reveals lattice *c* constraints during the H2-H3 transition. We concur with the reviewer that crystallite size may influence capacity retention and rate capabilities and view this as a compelling topic for future investigation. As discussed in the original manuscript: “Interestingly, the average domain size in the sample processed at 650°C, where the layered phase fraction reaches 87%, is 35 nm—larger than the 23 nm observed in the sample processed at 600°C. The larger primary particle size of LiNiO₂, especially with increased Ni/Li mixing, may contribute to a reduction in capacity. However, the size effects of LiNiO₂ in the (LiNiO₂)_L|(Li_xNi_{2-x}O₂)_{RS} composite on electrochemical performance remain to be thoroughly explored.”

The reviewer's comment raises a pertinent question about the relative importance of primary and secondary particle sizes and the interlocking dual-phase configuration in enhancing cycling stability. For instance, variations in secondary particle sizes in NMC cathodes can lead to differences in their cycling stability (Nature Communications, 2017, 8:14589, DOI: 10.1038/ncomms14589). In response, we have examined and compared the 600C-xh (where x = 3, 6, 12, 18, 36, and 48) samples as well as the 700C-6h sample using SEM images (Figure R3), including cross-sectional views (Figure R4). The SEM results indicate consistent secondary particle morphology across the 600C-xh samples, suggesting that sintering time has a negligible impact on secondary particle size and shape at 600°C. This demonstrates that the different cycling stabilities among 600C-xh samples are not due to their secondary particle size and shape.

However, cross-sectional images (Figure R4) reveal a trend of increasing internal porosity in the 600C-xh samples as heating time increases, likely due to the growth of primary particles. Although the effect of porosity on cycle stability for LNO cathodes requires further investigation, studies (Scientific Reports, 2017, 7:42521, DOI: 10.1038/srep42521, ACS Applied Energy Materials, 2021, 4, 1993-2003) suggest that porosity can enhance specific capacity, rate capability, and cycle stability by allowing electrolyte wetting within the internal pores, thereby shortening Li-ion diffusion paths and promoting homogeneous delithiation. This process results in more uniform stress distribution during cycling (Communications Materials, 2023, 4:90, <https://doi.org/10.1038/s43246-023-00418-8>). Furthermore, internal pores within secondary particles may accommodate strain generated during cycling, thereby improving cycling stability (Advanced Energy Materials, 2017, 7, 1602559). Therefore, we cannot attribute the observed decrease in cycle stability for the 600C-xh samples with extended heating times to the observed increase in porosity.

The crystalline sizes of both the rocksalt (RS) and layered components, as shown in Table S1, increase with extended heating times of 3, 6, and 12 hours at 600°C, all below 100 nm. Literature indicates that for nanoscale layered oxide cathodes, smaller crystallites are more susceptible to adverse surface reactions, leading to structural degradation and shorter cycling life (Communications Materials, 2023, 4:90, <https://doi.org/10.1038/s43246-023-00418-8>). For instance, LiCoO₂ cathodes prepared with primary particle sizes of 50, 100, and 300 nm show increased reversible capacity as particle size grows (Particuology, 2022, 61, 18-29). However, in our study, the trend is reversed: larger crystallites exhibit poorer cycling stability. Thus, for the dual-phase cathode with a significant rocksalt phase content (>7%, Table S2) acting as a structural stabilizer, performance is less influenced by crystal size. With prolonged heating, such as from 18 to 48 hours at 600°C, the reduction in the rocksalt phase diminishes its stabilizing effect, making cycle performance increasingly dependent on other factors like internal stress within micron-sized primary particles.

In summary, the size and morphology of the secondary particles, along with the crystallite sizes in the primary particles, do not significantly contribute to the enhanced cycling stability of the 600C-6h cathode. However, the primary crystallite size may influence the initial discharge capacity and rate capability (Nature Communications, 2024, 15:430). Effectively controlling both primary and secondary particle size and morphology in dual-phase cathodes remains an intriguing area for future research.

Figure R3 and R4 are added to the Supplementary Information as corresponding Figure S9 and S10. The following discussion has been added to the main text of the revised manuscript.

“In addition to examining the kinetic evolution of the structural and morphological parameters of the 600C-xh samples (**Table S1**), we also studied the kinetic effects on the secondary particles of the 600C-xh cathodes. As shown by the SEM images (**Figure S9**), the shape and size of the secondary particles remain largely unchanged, whereas the internal porosity shows a slight increase with extended heating times at 600°C (**Figure S10**). Drawing from literature reports on the influence of particle/crystal size^{21,22} and internal porosity²³⁻²⁵ on cycling stability, we confirm that the rocksalt phase fraction is the most critical factor affecting cycling stability.

.”

Fig. R3 | SEM images of different samples showing morphologies of secondary particles. a, 600C-3h. b, 600C-6h. c, 600C-12h. d, 600C-18h. e, 600C-36h. f, 600C-48h. g, 700C-6h.

Fig. R4 | SEM images of different samples showing cross-sectional views of secondary particles. a, 600C-3h. b, 600C-6h. c, 600C-12h. d, 600C-18h. e, 600C-36h. f, 600C-48h. g, 700C-6h.

3. To evaluate the effectiveness of the interlock configuration in the 600°C-6h composite, the authors synthesized s-LNO as a control. Although Rietveld refinement results were provided in Table 1, there was insufficient data to confirm whether this control was reasonable. First, please provide HAADF-STEM images and SAED patterns to confirm that the synthesized s-LNO was indeed a single phase. Second, please provide SEM images showing particle morphology and size. Lastly, include the raw data from the Rietveld refinement.

Response:

We thank the reviewer for their careful review and valuable comments. The material characterization of the s-LNO (700C-6h sample) is provided in Figure R5 below and has also been included as the new Figure S13 in the revised Supplementary Information (SI) to confirm its single-layered phase.

Figure R5(a) presents the SXRD pattern of s-LNO along with the calculated patterns from Rietveld refinement, providing strong evidence of its single-layered phase. Furthermore, the HAADF-STEM image with the FFT pattern (Figures R5(c-e)) further confirms the single-layered structure. The SEM image (Figure R5(b)) depicts the morphology and corresponding particle size.

In addition to incorporating Figure S13 in SI, we have added the following description to the main text of the manuscript.

“For comparison, structural and morphological characterizations of the baseline s-LNO are shown in **Figure S13**, with structural parameters summarized in **Table S1**. Additionally, its SEM images and first-cycle charge/discharge curves are presented in **Figures S9, S10, and S11**, respectively, to facilitate comparison with the 600C-xh cathodes.”

Fig. R5 | Material characterizations of the layered s-LNO, *i.e.*, the 700C-6h sample. **a**, SXRD pattern in comparison to the calculated patterns by Rietveld refinement. In the plots, blue circles are used for the observed data, red lines for the calculated data, the green bar for the Bragg peak positions, and black lines for the difference between the observed and calculated patterns. **b**, SEM image showing particle morphology and size. **c**, HAADF-STEM image of primary particles. **d**, local surface in an atomic resolution and **e**, corresponding FFT pattern transferred from the red dash-lined area in **d**.

4. The SXRD data in Figure 3b showed that the phase transition during the first charge and discharge was asymmetric. The phase transition occurred at high voltage during charging, while it occurred at relatively low voltage during discharging. Additionally, the voltage profiles for the first cycle were missing in Figure 2e. This suggested that phase transitions in the first cycle might differ from those in subsequent cycles. This reviewer requests that the authors provide the voltage profiles for the first cycle of the 600°C-6h sample. If the phase transition occurred differently in the first cycle compared to subsequent cycles, please illuminate the reason for this.

Response:

We thank the reviewer for their careful review and valuable comments. Indeed, during the operando SXR D measurements, we applied Constant-Current Constant-Voltage (CCCV) charging. Specifically, the two cathodes were charged to 4.8 V at a constant current of 0.1C, followed by constant-voltage charging at 4.8 V until the current decreased to 0.01C. SXR D patterns were recorded at 4.8 V under these conditions.

Figures 3a-d have been updated to include the voltage curves as a function of time, as shown here in Figure R6. These curves suggest that the phase transitions during the initial charge and discharge processes of both cathodes are mostly reversible, consistent with the dQ/dV profile presented in Figure 2d.

Additionally, the voltage profiles for the first cycle at 0.1C, also subjected to the CCCV charging strategy, have been added to Figure 2e (shown here in Figure R7). These profiles align with the cycling data at 2C previously shown in Figure 2e. All voltage profiles, regardless of current density, indicate the same phase transitions. Apparently, the higher charge-discharge capacity of the 0.1C cycle is due to the rate capability of the cell.

Fig. R6 | Revised figures of Figure 3a-c. **a**, heatmaps of operando SXR D patterns recorded in the first cycle of the two cathodes at 0.1 C in a wide voltage range of 2.7-4.8 V vs. Li⁺/Li. The H2-H3 phase transition region is highlighted in dashed lines. **b** and **c**, magnified pattern evolution of 003, 107, 108 and 2-10 reflections in high charging states during the first cycle of the two cathodes.

Fig. R7 | Revised figure of Figure 2e. Charge/discharge curves at different cycles during long-term cycling.

cycling of the 600C-6h cathode through the activation cycle at 0.1 C, followed by subsequent 2C cycles.

5. Figure 2f presented cycle test results for full cells assembled as pouch cells, tested with different cut-off voltages to assess stability under conditions similar to commercial use. However, the data was presented in terms of capacity retention, without providing the actual capacity. Different cut-off voltages would likely result in varying capacities, and it is necessary to show these differences in terms of actual capacity. Additionally, for the sample tested within the 2.7 V–4.6 V voltage range, the initial capacity retention exceeded 100%, indicating some form of activation or another cause in the early cycles. Please illuminate the reason for this observation.

Response:

We thank the reviewer for the insightful comments. We also agree with the reviewer's comment that different cut-off voltages indeed result in varying capacities of full pouch cells. The actual capacity of two pouch cells cycled at different voltage ranges, *i.e.*, 2.7-4.2 V and 2.7-4.6 V have been added to Figure 2f and presented below in Figure R8. We further agree with the reviewer's comment that when the full pouch cell was cycled at a wide voltage range of 2.7-4.6 V, it underwent an initial activation process, leading to the capacity increase during initial tens of cycles.

Fig. R8 | Revised Figure 2f. Cycling capacities of 600C-6h||graphite pouch cells at 1C in voltage ranges of 2.7-4.2 and 2.7-4.6 V, respectively, together with corresponding columbic efficiencies versus cycles.

6. Figure 2a showed cycle test results for samples synthesized with different sintering times at 600°C. While this data presented capacity changes over cycles, it did not show how the voltage profiles differed depending on sintering time. Please include information on how phase changes varied with sintering time, including corresponding voltage profile data.

Response:

We thank the reviewer for the valuable comments. The phase changes of 600C-xh samples dependent on the varied sintering time was summarized in Table S1 and presented as solid black spheres in Figure 2b, indicating the varied fraction of rocksalt phase from 2.2 to 20.1 wt.%. The charge and discharge curves of 600C-xh cathodes in the first cycle have been added as the below Figure R9 (the new Figure S11 in the revised SI), which reveal very similar voltage profiles that reflect electrochemical performance of lithium-active layered LiNiO_2 component with gradually increased capacities with extended sintering hours from 3 to 48h. The corresponding description has

been added in the revised manuscript and revised Supplementary Information.

Fig. R9 | Initial charge/discharge curves of 600C-xh cathodes at 0.33 C in voltage range of 2.7-4.6 V in comparison with that of single-phase layered 700C-6h cathode.

7. The manuscript claimed that interlocking with the rock-salt component in the cathode composite alleviated the mechanical anisotropy of the layered component after electrochemical cycling, which was supported by SXR D data. However, please provide TEM images with atomic resolution to directly and intuitively observe any changes at the boundary between the rock-salt and layered phases after cycling.

Response:

We thank the reviewer for the valuable comments. To address the concerns, we conducted postmortem *ex-situ* SXR D measurements with Rietveld refinements on cycled cathodes after 1 and 10 cycles. These analyses, added as Figure R10a-c (Figure S15 in revised SI), reveal a well-preserved dual-phase structure with a balanced phase fraction in the bulk of the cycled 600C-6h cathode, as detailed in Table R1.

Furthermore, we performed HAADF-STEM imaging with atomic resolution on the 600C-6h cathode after 1000 cycles within a voltage range of 2.7–4.8 V at 2C. This imaging, presented in Figure R10d (Figure S16 in revised SI), confirms the preservation of phase domains and boundaries.

The corresponding descriptions have been incorporated into the revised manuscript and Supplementary Information.

“To maintain high capacity retention during cycling, the d-LNO cathode must preserve its interlocked dual-phase crystal structure and morphology, as verified by post-cycle XRD measurements. **Figure S15** and **Table S3** present the SXR D patterns and refined structural parameters, respectively, for the d-LNO cathode before cycling and after 1 and 10 cycles. The RS fractions in these three samples range from 17% to 21%, showing no dependence on the number of cycles, likely due to the nonuniform RS distribution

in the pristine powder. The LNO component retains its stoichiometric order, while the Li content in the RS phase and the LNO domain size remain consistent across all samples. This demonstrates that the dual-phase structure of the d-LNO cathode is preserved after multiple cycles and is the key factor contributing to cycling stability.

The samples for post-cycle SXR D measurements were prepared by removing the cathode material, along with the aluminum current collector, from coin cells after a specific number of cycles. The cells were fabricated as described in the **Methods** section. SXR D patterns were collected with the cathode powder on the aluminum foil, following the procedure outlined in the **Methods** section. The 0-cycle sample was obtained from a cell that underwent no cycling. The 1-cycle and 10-cycle samples were pre-activated with two cycles at 0.1C, followed by 1 and 10 cycles, respectively, at a C/3 rate within a voltage range of 2.7–4.8 V.

Additionally, HAADF-STEM imaging with atomic resolution was performed on the d-LNO cathode after 1000 cycles, cycled between 2.7 and 4.8 V at a 2C rate (**Figure 2c**). As shown by the image in **Figure S16**, phase domains and boundaries are preserved even after prolonged cycling.”

”

Fig. R10 | SXR D patterns with Rietveld refinements from dual-phase d-LNO after **a**, 0 cycle, **b**, 1 cycle and **c**, 10 cycles. The violet, blue, and green curves show the contribution from the Al (current collector) and rocksalt and layered phase, respectively. The samples were charge-discharged at 0.1C for two cycles and C/3 for the rest of cycles, from 2.7 to 4.8 V. SXR D data was taken with the cathode powder on the Al foil, directly removed from the cycled coin cell. **d**, HAADF-STEM image of cycled 600C-6h cathode after 1000 cycles in a voltage range of 2.7-4.8 V at 2 C.

Table R1. Parameters of the dual phase structure of the 600C-6h sample after 0 (open), 1 and 10 cycles. The unit of lattice parameters a -LNO, c -LNO and a -RS is Å, ST-LNO and CS-LNO are microstrain (%) and crystalline size (nm), respectively, for LiNiO₂. Li in RS and Li in LNO are lithium occupancy in RS and LNO structure respectively.

Cycle	c/a	a -LNO	c -LNO	a -RS	Li in RS	ST-LNO	DS-LNO	Li in LNO	RS (%)	Rwp (%)
0	4.9296	2.8736(1)	14.1657(10)	4.0682(3)	0.330(9)	0.97(5)	13.4(4)	1.000(10)	20.7	3.29965
1	4.9405	2.8717(1)	14.1877(8)	4.0695(3)	0.350(10)	0.92(3)	14.3(3)	1.000(7)	17.2	4.93098
10	4.9357	2.8762(1)	14.1964(8)	4.0740(2)	0.363(7)	1.11(4)	14.3(3)	1.000(8)	21.4	3.97229

8. In Figures 3f-i, the formation of intergranular nanocracks in s-LNO and d-LNO was identified using 3D TXM imaging. However, differences in the size or distribution of pores between primary particles in pristine LNO could be a cause of this observation. To confirm that nanocrack formation in s-LNO and d-LNO was due to the elastic coupling between the layered and rock-salt phases, please demonstrate that the secondary particles in pristine s-LNO and d-LNO had similar morphology and porosity using 3D TXM images before cycling.

Response:

We thank the reviewer for the insightful comments and understand their concerns regarding the initial morphology's influence on nano-crack formation in cycled s-LNO and d-LNO materials.

Rather than using 3D TXM images, we captured SEM images of the secondary particles from pristine s-LNO and d-LNO cathodes, as well as cross-sectional views of FIB-thinned specimens,

shown in Figures R3(b), (g) and R4(b), (g). Both the pristine s-LNO (700C-6h) and d-LNO (600C-6h) samples exhibit similar particle sizes, with slightly larger porosity observed in s-LNO. However, this small difference in porosity is unlikely to significantly influence nano-crack formation.

Notably, previous studies (ACS Appl. Energy Mater. 2021, 4, 1993-2003; Adv. Energy Mater. 2017, 7, 1602559) suggest that internal pore spaces can act as buffers to accommodate strain during cycling, thereby mitigating nano-crack formation. In our case, however, the cathode with slightly larger porosity exhibits more severe nano-crack formation, indicating that preexisting porosity is not the primary factor.

Several prior studies (ACS Energy Lett. 2021, 6(2), 687-693; Adv. Funct. Mater. 2019, 29(18), 1900247; Nano Lett. 2022, 22(3), 1278-1286) have attributed detrimental mechanical fractures in NMC cathodes to anisotropic lattice variations caused by repeated lithium-ion extraction and insertion in layered domains during cycling. Based on this, we propose that the elastic coupling between the layered and RS phases, which reduces the mechanical anisotropy of the layered component, is the critical factor inhibiting nano-crack formation.

Following text has been added to the revised manuscript:

“Cross-sectional SEM images (**Figure S10**) reveal the presence of internal pores in pristine s-LNO and d-LNO, formed during the synthesis process. However, open nanocracks develop exclusively in the secondary particles of s-LNO. The slight differences in internal pores formed during synthesis between the two cathodes are unlikely to be the primary factor driving nanocrack formation.”

Copied Fig. R3 | SEM images of different samples showing morphologies of secondary particles. a, 600C-3h. b, 600C-6h. c, 600C-12h. d, 600C-18h. e, 600C-36h. f, 600C-48h. g, 700C-6h.

Copied Fig. R4 | SEM images of different samples showing cross-sectional views of secondary particles. a, 600C-3h. b, 600C-6h. c, 600C-12h. d, 600C-18h. e, 600C-36h. f, 600C-48h. g, 700C-6h.

9. Please add a section on Transmission X-ray Microscopy (TXM) in the Methods section.

Response: We thank the reviewer for the careful reviewing. The information for Transmission X-ray Microscopy (TXM) measurement has been added in the Methods section in the revised manuscript.

10. The Rock-salt fraction (%) data for the 600°C-3h sample in Figure 2b did not match the Weight Fraction data for the rock-salt phase in Table 1. Please correct this discrepancy.

Response: We thank the reviewer for the careful reviewing. The rocksalt fraction is 20.1 wt.% in the 600C-3h sample, which has been corrected in the Figure 2b and Table S1 in the revised manuscript.

Reviewer #3 (Remarks to the Author):

I appreciate the author's efforts in investigating the electrochemical performance of the mixed layered and rock-salt LiNiO_2 composite cathode. Showcasing a cobalt-free cathode with high energy and long cycle life. However, there are serious concerns regarding the thorough understanding of this intermediate phase. This study also follows up on the layered $\text{LiNi}_{0.95}\text{Mn}_{0.05}\text{O}_2$ and rock salt intermixed phases (<https://doi.org/10.1038/s41467-023-44583-3>), which raises questions about the novelty of this study. The title of the paper is attractive, but I didn't find it relevant. The authors didn't control any growth orientation of the particles or preserve original structure of LNO.

Response:

We appreciate the reviewer's positive comments on our work. In response to the reviewer's feedback, we have revised the manuscript and incorporated new experimental data to further support our main conclusions. We believe these updates address the major concerns raised by the reviewer, as detailed

below.

The current manuscript presents a unique dual-phase structure, $(\text{LiNiO}_2)_L|(\text{Li}_x\text{Ni}_{2-x}\text{O}_2)_{\text{RS}}$, identified as a metastable intermediate using *in-situ* SXR and synthesized as a kinetically stable material. This composite structure demonstrates superior cycling stability when employed as a cathode material in Li-ion batteries. With experimental evidence provided by STEM imaging and *operando* SXR, we trace the origin of the dual-phase structure and investigate its contribution to enhanced electrochemical performance.

An earlier version of this work was submitted to and reviewed by Nature Chemistry in May 2022 under the title “A New Lithium Nickel Oxide Cathode Material with a Composite Structure for High-Performance Li-ion Batteries” and was simultaneously made available as a preprint on Research Square (Research Square, DOI: 10.21203/rs.3.rs-1450650/v1, 2022). The manuscript was not published in Nature Chemistry and has now been transferred to Nature Communications. In the current version, we have reorganized the manuscript and added more experimental results, including sectional HAADF-STEM images, TXM images, and long-term cycling tests on pouch cells, to further support our findings on the interlocked dual-phase structure and its exceptional electrochemical performance.

As mentioned by the reviewer, a related study on $\text{LiNi}_{0.95}\text{Mn}_{0.05}\text{O}_2$ (Nat. Commun., 2024, Jan 10; 15(1):430), with overlapping senior authors and funding sources of this work, reported a similar dual-phase structure in a 5% Mn-substituted lithium nickel oxide. That study emphasized the role of Li deficiency in kinetically controlling the composite structure and its electrochemical performance and cited our preprint as a reference.

We emphasized the significance of the topotaxial nature of the dual-phase structure growth by highlighting the term “Topotaxially Grown Composite” in the title. A topotaxial transformation refers to a solid-state reaction where the crystallographic orientation of the product phase is closely related to that of the reactant phase. As described by Goodenough in their 1958 paper (J. Phys. Chem. Solids 1958, 5 (1), 107-116), the cation ordering of the layered phase develops along the $\langle 111 \rangle$ directions of the rocksalt cubic lattice, indicating that the transformation from rocksalt (RS) to the layered phase is a topotaxial process. While crystallographic studies of the rocksalt-to-layered phase transition provide lattice relationships between the phases (Fig. 1 in Goodenough's paper), direct observation of this topotaxial transformation requires the coexistence of the two phases within the same primary particle. The dual-phase configuration, a metastable intermediate absent in the thermodynamic equilibrium phase diagram of the Li-Ni-O system, can only be captured through *in-situ* studies.

Using *in-situ* SXR, we identified this dual-phase configuration. Sectional HAADF-STEM imaging allowed us to directly visualize the intergrown domains of the rocksalt and layered phases and establish their crystallographic orientation relationships. To our knowledge, this represents the first direct visualization of the topotaxial transformation between these phases. The HAADF-STEM images confirm that the coherent boundaries between the two phases originate from their topotaxial relationship (Figures 1c-g).

This topotaxial relationship persists across the temperature range of the miscibility gap, enabling kinetic tuning to achieve an optimized dual-phase configuration. Indeed, this optimization does not involve controlling the growth orientation of the layered phase but does preserve the near-stoichiometric composition in the layered phase.

1. Why are authors claiming it as a new nickelate cathode? The Pseudo phase diagrams (of $\text{Li}_2\text{O-O-NiO}$) in earlier studies mentioned the intermediate phases. Please refer to Matteo Bianchini et al., (<https://onlinelibrary.wiley.com/doi/10.1002/anie.201812472>).

Response:

We thank the reviewer for the insightful comments. Simply put, the dual-phase composite structure, consisting of a well-ordered layered phase and lithiated rocksalt phase, was not predicted in the phase diagram mentioned by the reviewer. Bianchini's paper was also cited in our manuscript (Ref. 4).

2. Is the same phase reproducible at that particular temperature and time with the same ratio of 13:87 of Rs-LNO and L-LNO? How many times did the authors test its performance, repeating from the synthesis to cycling performance? The growth kinetics of these intermediate phases are influenced by many factors; sometimes, they are unstable under ambient conditions and tend to form a stable phase or decompose. The growth orientation of these metastable intermediate phases can also depend on the crucibles used. Did the authors face any issues while handling dual-mix cathode?

Response:

We thank the reviewer for the valuable comments on the synthetic control and growth kinetics of the $(\text{LiNiO}_2)_L(\text{Li}_x\text{Ni}_{2-x}\text{O}_2)_{\text{RS}}$ composite. We didn't face any issue while reproducing our dual-phase samples with varied phase ratios of rocksalt and layered phases, and there is no growth orientation problem of obtained intermediate phases. The dual-phase samples were prepared at both Soochow University and Brookhaven National Laboratory independently with very consistent structure and property, which were also stable under ambient conditions for SXRD measurements and electrochemical tests with coin cells and full pouch cells. All 600C-xh samples with their assigned phase ratios, as well as the 700C-6h sample, were reproducible and controllable through precise adjustment of sintering temperatures and corresponding durations. However, the reviewer is right that crucibles with varying temperature calibrations and gas flow schemes may affect growth kinetics, therefore the nominal heating temperature and time required to achieve the dual-phase composite with optimized electrochemical properties could vary in different labs. Thus, we identified the rocksalt-to-layered phase ratio of 13:87 as a benchmark for optimized structural configuration, independent of specific heating temperature and duration. The label "600C-6h" should be regarded as a sample designation, reflecting the experimental conditions specific to the heating devices in our laboratories.

3. Does this mixed phase have any impact on the diffusion kinetics of the Li-ion migration? The authors didn't talk about kinetics.

Response:

We thank the reviewer for the insightful comments regarding the lithium diffusion kinetics of dual-phase $(\text{LiNiO}_2)_L|(\text{Li}_x\text{Ni}_{2-x}\text{O}_2)_{\text{RS}}$ composites. The RS fraction in dual-phase cathodes does indeed influence lithium diffusion kinetics. As shown in Figure R9 (Figure S11 in the revised SI), cathodes with shorter sintering times and higher RS fractions exhibit increased polarization at lower states of charge (SOC) due to the RS phase being inactive within the composite. However, at higher SOC, the stronger interlocking effect provided by the RS phase mitigates *c*-lattice shrinkage caused by the H2-H3 transition, thereby enhancing lithium mobility in cathodes with higher RS fractions.

As a result of the double-edged effect, the rate capability comparison between the 600C-6h and s-LNO cathodes, as shown in Figure R11 (Figure S12 in the revised SI), reveals no significant difference in the rate dependence of their overall specific capacities.

Figure R9 and R11 are added in the revised Supplementary Information, as Figure S11 and S12 respectively, with related discussion in the manuscript.

Copied Fig. R9 | Initial charge/discharge curves of 600C-xh cathodes at 0.33 C in voltage range of 2.7-4.6 V in comparison with that of single-phase layered 700C-6h cathode.

Fig. R11 | Cycling capacity of 600C-6h and s-LNO cathode at different rates. The cells were cycled in a voltage range of 2.7 – 4.8 V vs. Li⁺/Li.

4. This study is missing a performance comparison with reported literature. It seems the authors control S-LiNiO₂ is not up to standards. Compare with the best LNO perform from the literature.

Response:

We thank the reviewer for the valuable comments. In a recent study (Adv. Energy Mater. 2022, 12, 2103067), the electrochemical cycling performance of pure LNO and several tungsten-doped LNO samples was compared (Figure below). The control LNO exhibited a capacity retention of 82% after 50 cycles (from 170 mAh/g to 140 mAh/g) within a voltage range of 3.0-4.3 V at a rate of C/5. In another study (Energy Storage Materials 66 (2024) 103200), the capacity decreased from 220 mAh/g to 160 mAh/g over 30 cycles, corresponding to a 73% capacity retention, when cycled between 2.5 to 4.5 V at a very low rate (10 mA/g).

Copied Figure 5A from the literature (Adv. Energy Mater. 2022, 12, 2103067) | Electrochemical

cycling performance of LNO, W0.5-LNO, W1-LNO, W2-LNO, and W4-LNO synthesized at 800° C (2 cycles at C/20, 50 cycles at C/5, and 2 cycles at C/20).

In contrast, our s-LNO control cathode demonstrates superior performance, achieving 86% capacity retention (180 mAh/g from an initial 209 mAh/g) after 50 cycles at a wider voltage range (2.7-4.8 V) and a higher rate of 2 C, as shown in Fig. 2c. This highlights that the performance of our s-LNO control is comparable to, or better than, those reported in the literature.

Moreover, our dual-phase 600C-6h composite cathode reveals considerably improved cycling performance, achieving 87% capacity retention (167 mAh/g from an initial 191 mAh/g) after 1000 cycles in such a wider voltage range (2.7-4.8 V) at 2 C, as shown in Figure 2c. Following the reviewer's suggestion, a comprehensive list of LNO performance metrics from the literature is included in the revised SI. The newly-added Table S2 shows comparative specific capacity and capacity retention of the dual-phase 600C-6h and single-phase s-LNO cathodes obtained in this work compared to that of various LiNiO₂-based cathodes reported in the literature to date. The corresponding description has been added in the revised manuscript and revised Supplementary Information.

Table S2. Comparative specific capacity and capacity retention of the dual-phase 600C-6h and single-phase s-LNO cathodes obtained in this work compared to that of various LiNiO₂-based cathodes reported

Systems	Synthesis Strategy	Voltage Range (V)	Initial Capacity (mAh g ⁻¹)	Capacity retention (%)	Ref.
LiNiO ₂ Li _{0.8} Ni _{1.8} O (full-cell, 1C =180 mA g ⁻¹)	Composite Structure (600°C-6h)	2.7-4.2 V (vs. graphite)	2Ah pouch cell	Above 90% after 500 cycles	This work
LiNiO ₂ Li _{0.8} Ni _{1.8} O (full-cell, 1C =180 mA g ⁻¹)	Composite Structure (600°C-6h)	2.7-4.6 V (vs. graphite)	2Ah pouch cell	80% after 1000 cycles	This work
LiNiO ₂ Li _{0.8} Ni _{1.8} O (half-cell, 1C =180 mA g ⁻¹)	Composite Structure (600°C-6h)	2.7-4.8 V (vs. Li ⁺ /Li)	240.7 mAh g ⁻¹ (at 0.1C)	88% after 1000 cycles, at 2C	This work
LiNiO ₂ Li _{0.8} Ni _{1.8} O (half-cell, 1C =180 mA g ⁻¹)	Composite Structure (600°C-6h)	2.7-4.6 V (vs. Li ⁺ /Li)	231.5 mAh g ⁻¹ (at 0.1C)	100% after 50 cycles, at 0.333 C	This work
LiNiO ₂ Li _{0.8} Ni _{1.8} O (half-cell, 1C =180 mA g ⁻¹)	Composite Structure (600°C-6h)	2.7-4.4 V (vs. Li ⁺ /Li)	222.2 mAh g ⁻¹ (at 0.1C)	99.7% (after 200 cycles, at 0.5 C);	This work
LiNiO ₂	Pristine (700°C-6h)	2.7-4.4 V (vs. Li ⁺ /Li)	246.1 mAh g ⁻¹ (at 0.1C)	60.0% (after 200 cycles, at 0.5 C);	This work
LiNiO ₂	Pristine (700°C-6h)	2.7-4.8 V (vs. Li ⁺ /Li)	256.9 mAh g ⁻¹ (at 0.1C)	67.5% (after 200 cycles, at 0.5 C);	This work
LiNiO ₂ @Al&Mg (half-cell, 1C =200 mA g ⁻¹)	Mg&Al dual-doping (480°C-5h, then 720°C-12h)	2.8-4.6 V (vs. Li ⁺ /Li)	200 mAh g ⁻¹ (at 0.1C)	70% (after 500 cycles, at 1 C);	¹
LiNiO ₂ @Nb-based coating (half-cell, 1C =190 mA g ⁻¹)	Nb-based surface coating (400°C-4h, then 700°C-6h)	3.0-4.3 V (vs. Li ⁺ /Li)	217.5 mAh g ⁻¹ (at 0.1C)	84.3% (after 106 cycles, at 0.5 C);	²
LiNiO ₂	Pristine	2.8-4.4 V	245 mAh g ⁻¹ (at	79% (after 200 cycles,	³

(half-cell, 1C =200 mA g ⁻¹)	(550°C-5h, then 690°C-12h)	(vs. Li ⁺ /Li)	0.1C)	at 1 C);	
LiNiO ₂ (half-cell, 1C =220 mA g ⁻¹)	Pristine (480°C-2h, then 680°C-10h)	2.8-4.3 V (vs. Li ⁺ /Li)	239.7 mAh g ⁻¹ (at 0.1C)	43.4% (after 100 cycles, at 1 C);	4
LiNiO ₂ (half-cell, 1C =280 mA g ⁻¹)	Core-shell structure (Li/Ni ratio: 0.95/1; 700°C-12h)	2.8-4.3 V (vs. Li ⁺ /Li)	200 mAh g ⁻¹ (at 0.1C)	89% (after 100 cycles, at 1 C);	5
LiNiO ₂ (half-cell, 1C =200 mA g ⁻¹)	Single Crystal and LiF coating (620°C -10h, then 700°C-4h)	2.7-4.4 V (vs. Li ⁺ /Li)	234.5 mAh g ⁻¹ (at 0.1C)	65.1% (after 200 cycles, at 2 C);	6
LiNiO ₂ (full-cell 1C =180 mA g ⁻¹)	Nb-coating (500°C -5h, then 700°C-12h)	2.5-4.2 V (vs. graphite)	194 mAh g ⁻¹ (at 0.5C charge/1C discharge)	82% (after 200 cycles, at 0.5C charge/1C discharge);	7
LiNiO ₂ (full-cell 1C =180 mA g ⁻¹)	Electrolyte engineering (500°C -5h, then 700°C-12h)	2.8-4.3 V (vs. Li ⁺ /Li)	205 mAh g ⁻¹ (at 0.5C charge/1C discharge)	88% (after 250 cycles, at 0.5C charge/1C discharge);	8
LiNiO ₂ , pouch cell	No doping	2.6 – 4.3 V	190 mAhg ⁻¹ (at 0.2 C)	71% (after 400 cycles, at 0.2C)	9
LiNiO ₂ @Y (Half-cell,1C =180 mA g ⁻¹)	Y doping	2.8-4.3 V (vs. Li ⁺ /Li)	220 mAh g ⁻¹ (at 0.1C)	63.1% (after 100 cycles, at 0.5C)	10
LiNiO ₂ (Half-cell,1C =200 mA g ⁻¹)	Electrolyte design	2.8-4.4 V (vs. Li ⁺ /Li)	220 mAh g ⁻¹ (at 1/3C)	92% (after 200 cycles, at 0.5C charge/1C discharge)	11
LiNiO ₂ @ Graphene (Coin-type full cell, 1C =200 mA g ⁻¹)	Graphene coating	2.7-4.6 V (vs. graphite)	~190 mAh g ⁻¹ (at 1C)	76.1% (after 100 cycles, at 1C)	12
LiNiO ₂ (half-cell, 1C =180 mA g ⁻¹)	Pristine (200°C-5h, then 700°C-15h)	2.75-4.3 V (vs. Li ⁺ /Li)	235.2 mAh g ⁻¹ (at 0.1C)	80.2% (after 100 cycles, at 0.5C)	13
LiNiO ₂ (half-cell)	No doping	3.0- 4.3 V	200 mAhg ⁻¹ (C/20)	83% (after 50 cycles, 0.2C)	14
LiNiO ₂ (half-cell)	1% W doping	3.0-4.3 V	235 mAhg ⁻¹ (C/20)	89% (after 100 cycles, 0.2C)	14
LiNiO ₂ @Mg&Cu (pouch-type full cell, 1C =180 mA g ⁻¹)	Mg&Cu dual-doping (700°C-12h)	2.5-4.3 V (vs. MCMB)	230 mAh g ⁻¹ (at 0.1C)	81% (after 200 cycles, at 0.5C)	15
LiNiO ₂ (pouch-type full cell, 1C =180 mA g ⁻¹)	Tune oxygen pressure in synthesis (500°C-3h, then 685°C-12h)	2.5-4.2 V (vs. graphite)	181.8 mAh g ⁻¹ (at 1C)	76% (after 1000 cycles, at 1C)	16
LiNiO ₂ @Mn&Mg (half-cell, 1C =200 mA g ⁻¹)	Mn, Mg dual-doping (460°C-2h, then 700°C-6h)	2.7-4.4 V (vs. Li ⁺ /Li)	220 mAh g ⁻¹ (at 0.1C)	76% (after 350 cycles, at 0.5 C)	17
LiNiO ₂ @Ga	Ga doping	3-4.3 V	~205 mAh g ⁻¹	78% (after 100 cycles,	18

(Half-cell, 1C = 225 mA g ⁻¹)		(vs. Li ⁺ /Li)	(at 0.1C)	at 0.5C)	
LiNiO ₂ @Cu (Half-cell, 1C = 180 mA g ⁻¹)	Cu doping	2.5-4.5 V (vs. Li ⁺ /Li)	218 mAh g ⁻¹ (at 0.1C)	85% (after 100 cycles, at 0.5C)	19
LiNiO ₂ @Mg&Ti (Half-cell, 1C = 200 mA g ⁻¹)	Mg&Ti Co-doping	2.5-4.4 V (vs. Li ⁺ /Li)	208 mAh g ⁻¹ (at 0.1C)	85% (after 300 cycles, at 1C)	20
LiNiO ₂ @W (Half-cell, 1C = 180 mA g ⁻¹)	W doping	2.7-4.3 V (vs. Li ⁺ /Li)	236.1 mAh g ⁻¹ (at 0.1C)	90.1% (after 100 cycles, at 0.5C)	21
LiNiO ₂ (half-cell, 1C = 200 mA g ⁻¹)	Design in-situ-formed interphases	2.7-4.4 V (vs. Li ⁺ /Li)	265 mAh g ⁻¹ (at 1/15C)	81% (after 400 cycles, at 0.5 C)	22
LiNiO ₂ @2%Ti (half-cell, 1C = 180 mA g ⁻¹)	Ti doping (550°C-5h, then 690°C- 12h)	3-4.2 V (vs. Li ⁺ /Li)	211.3 mAh g ⁻¹ (at 0.1C)	85.2% (after 50 cycles, at 1C)	23
LiNiO ₂ (half-cell, 1C = 180 mA g ⁻¹)	Pristine (650°C-10h)	2.7-4.3 V (vs. Li ⁺ /Li)	246.6 mAh g ⁻¹ (at 0.1C)	75.2% (after 100 cycles, at 0.5C)	24
LiNiO ₂ @ 0.5%Zr (half-cell, 1C = 180 mA g ⁻¹)	Zr doping (650°C-10h)	2.7-4.3 V (vs. Li ⁺ /Li)	246.5 mAh g ⁻¹ (at 0.1C)	81% (after 100 cycles, at 0.5C)	25
LiNiO ₂ @ 1.4%Zr (half-cell, 1C = 180 mA g ⁻¹)	Zr doping (650°C-10h)	2.7-4.3 V (vs. Li ⁺ /Li)	232.6 mAh g ⁻¹ (at 0.1C)	86% (after 100 cycles, at 0.5C)	26
LiNiO ₂ @Co&Ti (half-cell, 1C = 200 mA g ⁻¹)	Co, Ti dual-doping	2.7-4.3 V (vs. Li ⁺ /Li)	214 mAh g ⁻¹ (at 0.1C)	98.7% (after 50 cycles, 0.5C charge/1C discharge)	27
LiNiO ₂ @Na (Half-cell, 1C = 270 mA g ⁻¹)	Na doping	3-4.4 V (vs. Li ⁺ /Li)	192 mAh g ⁻¹ (at 0.5C)	76% (after 100 cycles, at 0.5C)	28
LiNiO ₂ @Co&Mn (Half-cell, 1C = 180 mA g ⁻¹)	Co&Mn dual doping	2.7-4.3 V (vs. Li ⁺ /Li)	238 mAh g ⁻¹ (at 0.1C)	85% (after 100 cycles, at 0.5C)	29

References for Table S2

1. Zhou, J. *et al.* Mg/Al Double-Pillared LiNiO₂ as a Co-Free Ternary Cathode Material Ensuring Stable Cycling at 4.6 V. *ACS Applied Materials & Interfaces* **16**, 13948-13960 (2024).
2. Nunes, B. N., Karger, L., Zhang, R., Kondrakov, A. & Brezesinski, T. Enhanced Cycling Performance of the LiNiO₂ Cathode in Li-ion Batteries Enabled by Nb-based Surface Coating. *ChemSusChem*, e202402202 (2024).
3. Bai, Z. *et al.* Enabling High Stability of Co-Free LiNiO₂ Cathode via a Sulfide-Enriched Cathode Electrolyte Interface. *ACS Energy Letters* **9**, 2717-2726 (2024).
4. Yuwono, R. A. *et al.* Evaluation of LiNiO₂ with minimal cation mixing as a cathode for Li-ion batteries. *Chemical Engineering Journal* **456**, 141065 (2023).
5. Chen, J. *et al.* Constructing a thin disordered self-protective layer on the LiNiO₂ primary particles against oxygen release. *Advanced Functional Materials* **33**, 2211515 (2023).
6. Lee, D.-h. *et al.* Regulating Single-Crystal LiNiO₂ Size and Surface Coating toward a High-Capacity Cathode for Lithium-Ion Batteries. *ACS Applied Energy Materials* **6**, 5309-5317 (2023).
7. Ober, S., Mesnier, A. & Manthiram, A. Surface stabilization of cobalt-free LiNiO₂ with niobium

- for lithium-ion batteries. *ACS Applied Materials & Interfaces* **15**, 1442-1451 (2023).
8. Guo, Z., Cui, Z., Sim, R. & Manthiram, A. Localized High-Concentration Electrolytes with Low-Cost Diluents Compatible with Both Cobalt-Free LiNiO₂ Cathode and Lithium-Metal Anode. *Small* **19**, 2305055 (2023).
 9. Park, K. Y. *et al.* Elucidating and mitigating high-voltage degradation cascades in cobalt-free LiNiO₂ lithium-ion battery cathodes. *Advanced Materials* **34**, 2106402 (2022).
 10. Zhang, Y. *et al.* Enhancing LiNiO₂ cathode materials by concentration-gradient yttrium modification for rechargeable lithium-ion batteries. *J. Energy Chem.* (2021).
 11. Langdon, J., Cui, Z. & Manthiram, A. Role of Electrolyte in Overcoming the Challenges of LiNiO₂ Cathode in Lithium Batteries. *ACS Energy Lett.* **6**, 3809-3816 (2021).
 12. Park, K. Y. *et al.* Elucidating and Mitigating High-Voltage Degradation Cascades in Cobalt-Free LiNiO₂ Lithium-Ion Battery Cathodes. *Adv. Mater.*, 2106402 (2021).
 13. Ji, H. *et al.* Electrolyzed Ni(OH)₂ Precursor Sintered with LiOH/LiNiO₃ Mixed Salt for Structurally and Electrochemically Stable Cobalt-Free LiNiO₂ Cathode Materials. *ACS Appl. Mater. Interfaces* **13**, 50965-50974 (2021).
 14. Kitsche, D. *et al.* The effect of gallium substitution on the structure and electrochemical performance of LiNiO₂ in lithium-ion batteries. *Materials Advances* **1**, 639-647 (2020).
 15. Seong, W. M. & Manthiram, A. Complementary Effects of Mg and Cu Incorporation in Stabilizing the Cobalt-Free LiNiO₂ Cathode for Lithium-Ion Batteries. *ACS Appl. Mater. Interfaces* **12**, 43653-43664 (2020).
 16. Mesnier, A. & Manthiram, A. Synthesis of LiNiO₂ at Moderate Oxygen Pressure and Long-Term Cyclability in Lithium-Ion Full Cells. *ACS Appl. Mater. Interfaces* **12**, 52826-52835 (2020).
 17. Mu, L. *et al.* Structural and electrochemical impacts of Mg/Mn dual dopants on the LiNiO₂ cathode in Li-metal batteries. *ACS Appl. Mater. Interfaces* **12**, 12874-12882 (2020).
 18. Kitsche, D. *et al.* The effect of gallium substitution on the structure and electrochemical performance of LiNiO₂ in lithium-ion batteries. *Mater. Adv.* **1**, 639-647 (2020).
 19. Kong, X.-Z., Li, D.-L., Lahtinen, K., Kallio, T. & Ren, X.-Q. Effect of Copper-Doping on LiNiO₂ Positive Electrode for Lithium-Ion Batteries. *J. Electrochem. Soc.* **167**, 140545 (2020).
 20. Mu, L. *et al.* Dopant distribution in Co-free high-energy layered cathode materials. *Chem. Mater.* **31**, 9769-9776 (2019).
 21. Ryu, H.-H., Park, G.-T., Yoon, C. S. & Sun, Y.-K. Suppressing detrimental phase transitions via tungsten doping of LiNiO₂ cathode for next-generation lithium-ion batteries. *J. Mater. Chem. A* **7**, 18580-18588 (2019).
 22. Deng, T. *et al.* Designing in-situ-formed interphases enables highly reversible cobalt-free LiNiO₂ cathode for Li-ion and Li-metal batteries. *Joule* **3**, 2550-2564 (2019).
 23. Deng, S. *et al.* Structure and primary particle double-tuning by trace nano-TiO₂ for a high-performance LiNiO₂ cathode material. *Sustainable Energy Fuels* **3**, 3234-3243 (2019).
 24. Yoon, C. S., Jun, D.-W., Myung, S.-T. & Sun, Y.-K. Structural stability of LiNiO₂ cycled above 4.2 V. *ACS energy letters* **2**, 1150-1155 (2017).
 25. Yoon, C. S. *et al.* Cation ordering of Zr-doped LiNiO₂ cathode for lithium-ion batteries. *Chem. Mater.* **30**, 1808-1814 (2018).
 26. Yoon, C. S. *et al.* Self-passivation of a LiNiO₂ cathode for a lithium-ion battery through Zr doping. *ACS Energy Lett.* **3**, 1634-1639 (2018).

27. Ko, H. S., Kim, J. H., Wang, J. & Lee, J. D. Co/Ti co-substituted layered LiNiO₂ prepared using a concentration gradient method as an effective cathode material for Li-ion batteries. *Journal of Power Sources* **372**, 107-115 (2017).
28. Kim, H. *et al.* Role of Na⁺ in the cation disorder of [Li_{1-x}Na_x]NiO₂ as a cathode for lithium-ion batteries. *J. Electrochem. Soc.* **165**, A201 (2018).
29. Yoon, C. S. *et al.* Extracting maximum capacity from Ni-rich Li[Ni_{0.95}Co_{0.025}Mn_{0.025}]O₂ cathodes for high-energy-density lithium-ion batteries. *J. Mater. Chem. A* **6**, 4126-4132 (2018).

5. The authors claim that interlocking LiNiO₂ with rock salt suppress the severity of the H₂ to H₃ phase transition. However, there is still, a considerable change in lattice parameter *c* for d-LNO = 0.8 and s-LNO = 1 Å. Surprisingly, TXM doesn't show any interparticle cracking for Rs-LNO. The authors should elaborate on their explanation. How can interlocking change the isotropic or anisotropic expansion of the layered portion of the particles (which is 87%)?

Response:

We thank the reviewer for the insightful comments. As described in our manuscript, within the same voltage range, the lattice parameter *c* in s-LNO changes from 14.41 Å to 13.41 Å, corresponding to a 1.0 Å difference, while in d-LNO, it changes from 14.22 Å to 13.59 Å, a 0.63 Å difference. This demonstrates that the elastic coupling between the rocksalt and layered phases significantly suppresses lattice shrinkage.

More importantly, the *c* parameter in d-LNO varies continuously as shown in Figure 3d, indicating elastic deformation, while in s-LNO, the lattice yields to delithiation-induced stress, resulting in an abrupt lattice change around 4.3 V. As stated in our manuscript: "The substantial lattice mismatches in s-LNO between H₂ and H₃ phases, evidenced by the 003 peak splitting at 4.3 V (Figure 3d, inset), lead to a high misfit dislocation density, generating fresh and reactive surfaces, accompanied by lattice oxygen release and other irreversible reactions. In contrast, the sudden lattice collapse is avoided in d-LNO due to the elastic coupling between its rocksalt and layered domains (Figure 1e), which suppresses the accompanying irreversible reactions and greatly enhances structural stability during electrochemical cycles." Therefore, the anisotropic plastic deformation of the layered phase within the dual-phase d-LNO sample is inhibited by the interlock effect, mitigating the formation of nanocracks in their secondary particles. This preservation of the intergranular structure in the secondary d-LNO particles was confirmed by the 3D TXM imaging as shown in Figure 3h-i.

The interlocking effect of the 13% RS phase with the 87% layered phase is more intuitively understandable compared to other reported methods, where the stabilizing components are present in much smaller amounts. For example, in one study, 2.1% Nb-coating/substitution in NMC9055 was sufficient to modify the structure and improve long-term cycling stability (Chem. Mater., 2022, 34, 7858–7866). Similarly, introducing only 0.82 wt% of La₄(LiTM)O₈ as a "rivet" in NMC cathodes significantly reduced lattice strain during electrochemical cycles (Nature, Vol 611, 3 November 2022). As noted in our manuscript, the interlocking effect in the dual-phase composite is not limited to the RS-LNO domain boundaries but is further enhanced by coherent twin boundaries within the layered domains. These boundaries introduce additional coupling effects that

help mitigate anisotropic structural changes. Therefore, the 13% RS phase is sufficient to stabilize the 87% layered phase, as demonstrated in our study.

6. Why are the authors experiencing a huge IR drop in the cycling, especially during the discharge it is around 500 mV, as shown in Figure 2e?

Response

We thank the reviewer for the valuable comments. It is important to note that Figure 2e shows the cycling performance of the 600°C-6h cathode at a high current density of 2 C within a wide voltage range of 2.7–4.8 V vs. Li⁺/Li. Under such conditions, an inevitable electrochemical polarization, manifested as a significant IR drop, can occur, particularly in the half-cell testing system during charge/discharge. This behavior arises from the inherently high resistance of the coin cell configuration. A similar phenomenon has been reported in the literature (Nature Energy, 2021, 6, 495-505), where an extended voltage range of 3.0-4.7 V vs. Li⁺/Li at a relatively lower current density of 0.5 C also exhibited polarization effects, as illustrated in the figure below.

Copied Figure 2d from the literature (Nature Energy, 2021, 6, 495-505) | Voltage profiles of Li || NMC811 cells using 1 M LiPF₆/EC-EMC + 2% VC, showing apparent IR drops during the cycling.

7. Did the authors use O₂ gas for the synthesis of LNO? Generally, oxides are synthesized under oxygen flow to obtain good quality cathodes.

Response:

We greatly appreciate the reviewer's comment and fully agree. Oxygen (O₂) gas was indeed used during the synthesis of both the dual-phase composite and the single-phase LNO materials. The details of the sample preparation are provided in the Methods section of the revised manuscript.

8. Space group symbols are not written properly; the letter should be in italics.

Response:

We thank the reviewer for the careful reviewing. All the space group symbols have been corrected in italics in the revised manuscript.

9. The authors mentioned that “even moderate Ni mixing in the Li-Layer will deteriorate capacity retention”. The reported material contains 87% of layered structure with small portion of rock-salt. How is long-term cycling stability protected in this material? How does this rock salt structure protects anti-site mixing in the layered structure over the 1000 cycles? There is reason if dopants helping in stabilizing the structure, but I couldn't see relevance with interlocking.

Response:

We thank the reviewer for the insightful comments. The fact that LiNiO₂ with even moderate Ni mixing in the Li layer deteriorates capacity retention is well-established and supported by literature, such as Solid State Ionics 2001, 140 (1), 1-17 (Ref. 32 in the original manuscript). Additionally, the statement "The reported material contains 87% of layered structure with a small portion of rocksalt" is not clear regarding the composite structure. The dual-phase composite contains 13% rocksalt phase and 87% layered phase, with no observable anti-site mixing in the latter. These two phases exist in separate domains, sharing coherent boundaries, as confirmed by cross-sectional HAADF-STEM images (Figure 1c). The lattices of the two crystalline phases are well-aligned across the boundaries, demonstrated by the low coincidence lattice mismatch calculated from SXR measurements. This low lattice mismatch signifies strong elastic coupling between the phases, a critical factor in stabilizing the d-LNO cathode during electrochemical cycling.

Operando SXR analysis further supports this, showing that the elastic coupling in the dual-phase d-LNO cathode effectively suppresses lattice distortion. This is evidenced by the limited and continuous lattice variation during the H2 to H3 phase transition (Figure 3d). In contrast, the s-LNO cathode, which lacks the rocksalt phase's elastic coupling, yields to stress caused by structural shrinkage during delithiation. This results in the splitting into two phases (H2 and H3) with different lattice parameters, leading to high misfit dislocation density, the formation of fresh reactive surfaces, lattice oxygen release, and other irreversible reactions that significantly shorten cycling life. The interlocking mechanism in the dual-phase d-LNO cathode provides a unique structure-stabilizing approach, distinct from the stabilization achieved through doping strategies.

10. The authors mentioned that LNO started nucleating in the rock salt phase with different orientations. Can authors elaborate on this? What is the nucleation plan?

Response:

We thank the reviewer for the valuable comments. The layer ordering of LiNiO₂ (LNO), where Li and Ni occupy alternating (111) planes of the parent rocksalt (RS) crystal, was first described in Goodenough's seminal work (J. Phys. Chem. Solids 1958, 5 (1), 107-116, Ref. 16 in our manuscript).

Since there are four equivalent <111> cubic directions, the layered crystalline phase nucleates with equal probability along any of these directions. This was established by J. Dahn et al. (Phys. Rev. B 1992, 46 (6), 3236, Ref. 15 in our manuscript). To quote from their work: "As order develops, we do not know a priori along which of the four equivalent cubic directions [1 1 1], [1 -1 1], [1 1 -1],

or [1 -1 -1] the order develops. ... In the ordered state, only one of the four order parameters is significantly different from zero." In the appendix of their paper, Dahn et al. demonstrated that entropy is maximized when only one of the four order parameters is selected in their lattice-gas model.

In our study, the HAADF-STEM image in Figure 1c shows that ordering along different $\langle 111 \rangle$ directions can coexist within the same crystal. However, within each local domain, only one orientation is adopted to minimize the free energy. As stated in our manuscript: "Even though the layered crystalline nucleates with equal probability along the four $\langle 111 \rangle_{RS}$ orientations, only one of the directions is assumed in each isolated domain to minimize the free energy."

Reviewer #4 (Remarks to the Author):

The manuscript from Yao and Bai et al. presented a new explanation of the reaction mechanism of LiNiO_2 synthesized at low temperatures and at the short period of holding time. In the manuscript, the authors suggest a composite structure instead of a solid solution is formed at a non-equilibrium synthesis state. And the author tried to use some state-of-the-art analytics tools to resolve the mystery behind this. I believe this work is meaningful from academic perspective, but not from an industry perspective, given the lack of proper comparison due to lacking of care design of some control experiments, I would consider this manuscript potentially valuable but the authors need to make major revisions on many points to make this work meet the standard of Nature Communications. I recommend the authors to consider the following points:

1. It is difficult to distinguish if the material synthesized at 600°C -6h is a single phase or composite structure, one of the possible indicators is to measure and perform an in-depth analysis on the XRD pattern. Particularly the authors should focus on the "asymmetry" of the XRD peaks, please include this in your analysis.

Response:

We thank the reviewer for their valuable comments and suggestions regarding the structural analysis of the 600°C -6h cathode. We agree that more detailed XRD analysis is essential, and in response, we have added a section titled "Notes on the SXRD Data Analysis" in the Supplementary Information to provide a more comprehensive explanation of the structural analysis.

The overlap between the Bragg peaks of the rocksalt and layered phases does indeed make it challenging to identify the dual-phase composite. However, by carefully analyzing the peak profiles and relative intensities, and tracking the phase evolution trend in *in-situ* studies, we were able to unequivocally determine the dual-phase structure. The ability to correctly identify and further optimize this dual-phase structure with the aid of quantitative analysis of *in-situ* SXRD data highlights one of the novelties in this work. In the previous version of the paper, we did not provide an extensive discussion of the SXRD analysis. Instead, we referred to the derivation of the dual-phase model in our preprint available on Research Square (Ref. 30), as the cross-sectional HAADF-STEM image in Figure 1c already provided strong support for the dual-phase model. In response to

the reviewer's comments, we have now added a section titled "Notes on the SXRD Data Analysis" in the Supplementary Information to elaborate on the details of the SXRD structural analysis.

As shown in Figures 4a and 4b, the rocksalt phase, *i.e.*, $(\text{Li}_x\text{Ni}_{2-x}\text{O}_2)_{\text{RS}}$ formed first upon decomposition of the hydroxide precursor, followed by the emergence of the layered phase, *i.e.*, $(\text{LiNiO}_2)_{\text{L}}$ at around 400°C, as indicated by the appearance of the 003_L peak. At this stage, the layered phase exhibited weaker and broader peaks compared to the rocksalt phase, making the dual-phase model the most suitable representation of the material (refer to Figure S4 in the revised SI and its corresponding discussions, and included here as copied Figure R1).

Copied Fig. R1 | Three structural models used in the Rietveld refinement with respect to the SXRD pattern recorded at 483 °C during *in-situ* measurements. a, the layered structure with a $R\bar{3}m$ space group. b, the layered structure with a $R\bar{3}m$ space group and hkl dependent anisotropic peak broadening. The broad (003) peak fits better but the fitting for higher angle gets worse. c, the dual-phase model composing of the layered phase with a $R\bar{3}m$ space group and the Li-containing disordered rocksalt phase with a $Fm\bar{3}m$ space group, revealing the best fitting result compared to the other two models presented in a and b. In the plots, blue circles are used for the observed data, red lines for the calculated data, colored (green, navy and orange) bars for the Bragg peak positions, and black lines for the difference between the observed and calculated patterns.

At higher temperatures and with extended sintering times, the layered phase becomes the dominant component of the composite. However, the dual-phase model remains applicable for describing the Li-Ni-O system. This is supported by detailed X-ray crystallographic data, now elaborated in the "Notes on the SXRD Data Analysis" section of the revised SI, as well as HAADF-STEM images (Figures 1c and S1).

2. To follow up on question 1, for the Rietveld refinement in Fig. 1a, how was the reference of the

rock-salt phase $\text{Li}_x\text{Ni}_{2-x}\text{O}_2$ chosen, what is the x value, because it will impact the lattice parameters and therefore the ratio of this rock-salt phase.

Response:

We thank the reviewer for the valuable comments. The value of x in the RS phase ($\text{Li}_x\text{Ni}_{2-x}\text{O}_2$) was determined through refinements. Although SXR D is not sensitive to Li ions, the Li occupancy can be determined based on the Ni occupancy under the assumption that all metal sites are fully occupied by either Li or Ni ions. This approach has been applied in the literature, for example, in W. Li et al., Phys. Rev. B, 46, 3236 (1992).

The rocksalt phase emerges as the nickel hydroxide precursor decomposes and becomes lithiated during its early crystallization stages. As shown in Figure R12 (Figure S1 in the revised SI), prior to the formation of the layered phase, the predominant coexisting phases—besides minor phases like Li_2CO_3 —are $\text{Ni}(\text{OH})_2$ ($P\bar{3}m1$), lithiated rocksalt oxide ($Fm\bar{3}m$), and layered lithium nickel oxide ($R\bar{3}m$). At lower temperatures, the rocksalt peaks are broad and weak, reflecting the early stages of crystal nucleation and growth, and become sharper and more intense with increasing temperature, indicating continued crystal growth. The $(111)_{\text{RS}}$ peak, which aligns with the crystal direction of the subsequently formed layered phase, shifts to a higher angle as temperature increases. This shift corresponds to lattice contraction induced by lithiation and is consistent with the increasing x value in the rocksalt phase, as shown in Figure 4c of the main text.

Fig. R12 | a, Deconvoluted SXR D pattern based on the refinement analysis shows the co-existence of $\text{Ni}(\text{OH})_2$, $(\text{Li}_x\text{Ni}_{2-x}\text{O}_2)_{\text{RS}}$, LiOH , and Li_2CO_3 at low temperature ($T = 261 \text{ }^\circ\text{C}$). **b,** Enlarged SXR D patterns of three selected 2θ ranges recorded between 176 and 313 $^\circ\text{C}$ during *in-situ* measurements as shown in **Figure 4a** in the main text, showing the phase transformation from the $\text{Ni}(\text{OH})_2$ to the

lithiated disordered rock-salt phase. The P and RS are representatives for the $\text{Ni}(\text{OH})_2$ precursor and lithium-containing rocksalt phase, respectively.

3. Figures 1c-g show the HAADF-STEM and FFTs images of this composite material, however, one of the drawbacks of using this kind of high-resolution microscopy analytics method is it can only show the composition of the selected of interest within a very small area, as indicated by the scale bar of these images. My concerns regarding these images are, first, how was the images chosen to display. Second, I believe that for LiNiO_2 with even the least amount of cation mixing, you would still observe rock-salt layers, I agree the mass ratio of the rocksalt phase should be way less than 13%, but do they perform any difference in terms of the mechanism of providing structural and electrochemical stability? Please perform a similar analysis and add an in-depth discussion on “normal” LiNiO_2 as a comparison.

Response:

We thank the reviewer for the careful reviewing and valuable comments. To provide further details sample preparation for TEM imaging, we include a HAADF-STEM image of a sliced primary particle (Figure R2a) and the corresponding high-resolution image from a selected region (highlighted by the yellow square) inside the particle (Figure R2b). Notably, in Figure R2b, the white dashed line indicates the area where the image in Figure 1c was captured, clearly demonstrating that the rocksalt phase is distributed within the bulk structure of the primary particle. The reviewer is correct that HAADF-STEM imaging can only provide compositional information for a very small, selected area of interest. In this work, the statistical evaluation of the dual-phase composition was determined through quantitative analysis based on XRD measurements. The intergrown domains shown in Figure 1c are certainly not isolated exceptions, as similar features were observed in other samples. For instance, Figure S3a demonstrates a comparable dual-phase configuration in samples sintered at 350°C for 6 hours under an oxygen flow, where layered domains are just beginning to form within the RS crystal.

As shown in Figure 2a, the cycling stability is strongly influenced by the rocksalt/layered phase ratio. A higher rocksalt fraction in the cathode correlates with improved cycling stability. For instance, the 600C-36h cathode, with a rocksalt fraction of less than 5%, exhibits much poorer cycling stability compared to the 600C-6h cathode.

The SXRD pattern of s-LNO (Figure R5a), which reflects the bulk properties of the material, reveals less than 0.5% Ni mixing in the Li layer. At such a low level of cation mixing, no RS phase can be detected. Consequently, the interlock mechanism observed in the dual-phase d-LNO is absent in s-LNO. This is further confirmed by HAADF-STEM images of s-LNO (Figure R5c, d, and e), which clearly demonstrate the single-phase nature of the s-LNO samples.

Figure R5 is included in revised Supplementary Information as Figure S13 with related discussion.

Copied Fig. R2 | HAADF-STEM images of FIB-thinned $(\text{LiNiO}_2)_L|(\text{Li}_x\text{Ni}_{2-x}\text{O}_2)_{RS}$ composite obtained by sintering at 600 °C for 6 h. a, STEM image of a sliced primary particle prepared by the FIB. b, HAADF-STEM images captured in a selected region of the primary particle as marked by the white arrow and tallow square in a. The lined area in b is selected to show the typical interlocked rocksalt and layered phases in such as a composite material as presented in Figure 1c.

Copied Fig. R5 | Material characterizations of the layered s-LNO, *i.e.*, the 700C-6h sample. a, SXR D pattern in comparison to the calculated patterns by Rietveld refinement. In the plots, blue circles are used for the observed data, red lines for the calculated data, the green bar for the Bragg peak positions, and black lines for the difference between the observed and calculated patterns. b, SEM image showing particle morphology and size. c, HAADF-STEM image of primary particles. d, local surface in an atomic resolution and e, corresponding FFT pattern transferred from the red dash-lined area in d.

4. The Voltage–capacity curves of LNO-600°C 3-48h can be added to supplementary documents to compare the difference and reason behind this.

Response:

We thank the reviewer for the valuable comments. Per the reviewer's suggestion, we have included the charge and discharge curves of the 600C-xh cathodes during the first cycle, presented as Figure R9 below (and as the new Figure S11 in the revised Supplementary Information). These voltage profiles demonstrate progressively increasing capacities with extended sintering times, which can be attributed to the increasing fraction of the layered phase as the active cathode component (Figure 2b).

Interestingly, cathodes with longer sintering times exhibit higher polarization, manifested by steeper voltage slopes at high state of charge (SOC), compared to those sintered for shorter durations, such as the 600C-6h cathode. This behavior is an intrinsic property of single-phase LNO cathodes, where severe c-lattice shrinkage during and after the H2-H3 transition significantly narrows the Li-slab, making the deintercalation of residual Li-ions considerably more difficult at higher SOC (Phys. Rev. B 74, 094105, 2006, Small 2023, 19, 2300616).

In contrast, cathodes with shorter sintering times—and consequently a higher RS phase fraction, which acts as a structural stabilizer—mitigate c-lattice shrinkage, thereby reducing polarization at higher SOC and enabling additional capacity above 4.3 V. However, as a tradeoff, the cathodes with higher RS fractions display slightly increased polarization in the lower voltage range, as shown in the charge/discharge curves, resulting in a sacrifice in overall rate capability.

It is worth noting that, although the full capacity of the dual-phase cathode decreases with increasing sintering time and higher RS fraction, the degree of delithiation of the layered phase—the active component in the dual-phase cathodes—remains similar. This becomes evident when the specific capacities of cathodes with different RS fractions are normalized by the percentage of their active component. Therefore, the improved cycling stability observed in dual-phase cathodes with higher RS fractions is attributed to the stronger interlocking effect, as the lithium utilization in the active layered phase is independent of the RS fraction.

The 600C-6h cathode, with its optimized RS phase fraction, strikes a balance between high-voltage capacity and rate capability. More importantly, it demonstrates enhanced long-term cycling stability due to its stabilized structure.

Figure R9 is included in Supplementary Information as Figure S11 with related discussion.

Copied Fig. R9 Initial charge/discharge curves of 600C-xh cathodes at 0.33 C in voltage range of 2.7-4.6 V in comparison with that of single-phase layered 700C-6h cathode.

5. The cycling stability of LiNiO_2 is heavily determined by the amount of Li (x in $\text{Li}_{1-x}\text{NiO}_2$) being utilized, please add long-term cycling comparison with LiNiO_2 at 700°C-6h that only utilizes the capacity (low voltage) to the same level as composite cathode made at 600°C-6h (~195 mAh/g). Do the authors still expect a similar conclusion that this composite structure material will still outperform its competitor?

Response:

We thank the reviewer for the valuable comments. It is true that in general, cycling stability improves with reduced lithium utilization, as lower voltage limits the degree of lithiation, resulting in better stability. However, in the case of LiNiO_2 (LNO) cathodes, when the cutoff voltage exceeds 4.3 V, a structural phase transition from H2 to H3 occurs, causing severe lattice contraction, causing structural degradation, and low capacity retention over extended cycles. The unique interlocked dual-phase structure of the 600C-6h sample significantly mitigates structural deformation during the H2–H3 phase transition, enabling improved cycling stability at high voltage capacities. This enhanced stability cannot be attributed solely to reduced Li-ion utilization. Sun et al. investigated the dependence of cycling stability on cutoff voltage in single-phase LNO cathodes (DOI: 10.1021/acscenergylett.7b00304). They measured capacity retention in s-LNO cathodes with cutoff voltages of 4.3 V and 4.2 V, corresponding to $x = 0.93$ and 0.86 , respectively. The initial capacity at a 4.2 V cutoff and 0.5C rate was approximately 190 mAh/g, comparable to that of the 600C-6h cathode charged to 4.8 V at a 2C rate (191.2 mAh/g). Yet, the capacity retention in their s-LNO cathode was markedly lower (81% after 100 cycles) compared to 88% after 1000 cycles in our 600C-6h cathode. Therefore, we conclude that the enhanced cycling stability of our 600C-6h cathode is not mainly due to a lower initial discharge capacity, as seen in comparison with the 700C-6h cathode.

6. In Figure 2c, d, it is not fair to test s-LNO with an upper cut-off voltage of 4.8V, there is virtually no capacity above 4.3V for s-LNO and it only accelerates the parasitic reaction between s-LNO and electrolyte.

Response:

We thank the reviewer for the valuable comments. As followed by the reviewer's suggestion, s-LNO and d-LNO cathodes were cycled in narrow in a narrow voltage range of 2.7-4.4 V, and corresponding cycling performance of two different cathodes *i.e.*, the s-LNO(700C-6h) and 600C-6h (d-LNO) is shown as below Figure R13 (the new Figure S14 in the revised Supplementary Information), which reveals the similar conclusion in a wide 2.7-4.8 V that the dual-phase 600C-6h cathode has a better cycling stability compared to the single-phase 700C-6h cathode but with a lower initial capacity, owing to the structural integration of lithium inactive rocksalt phase. On the other hand, the high voltage capability of the d-LNO cathode enables higher capacity and is clearly an advantage of the new cathode.

Figure R13 is included in revised Supplementary Information as Figure S14 with related discussion.

Fig. R13 | Cycling performance up on 200 cycles of the dual-phase 600C-6h cathode compared to that of the single-phase s-LNO at 0.5 C in two different voltage ranges of 2.7-4.4 and 2.7-4.8 V vs. Li⁺/Li, respectively.

Reviewer #1 (Remarks to the Author):

I have carefully reviewed the revised manuscript and the authors' responses to the comments from all four reviewers. Overall, the authors have addressed the feedback in a reasonably comprehensive manner. However, as highlighted in Reviewer #3's Comment #2, the reproducibility of the results under the specified temperature, duration, and Rs-LNO to L-LNO ratio of 13:87 remains a concern. While the authors provide a definitive verbal assertion, such certainty is impractical given the complexity of sintering parameters. To substantiate this claim, the authors should present supporting evidence. Although the revised manuscript is generally reasonable, considering the high standards of *Nature Communications*, I am hesitant to recommend acceptance at this stage.

We sincerely appreciate the reviewer's careful assessment of our manuscript and responses. It is standard practice to report sintering conditions by specifying the heating temperature and duration, as being able to control these parameters is fundamental for maintaining proper quality assurance and control in any laboratory. In Table S2 of our manuscript, and thereby cited references, various sintering temperatures and durations are reported by many researchers for LNO cathode materials. However, we do not see any concern regarding the reproducibility of their synthesis, particularly at the laboratory scale.

Contrary to the reviewer's suggestion, the sintering parameters we adopted are not particularly complex compared to those reported in the literature. For example, our 600C-6h sample was synthesized by sintering a mixture of $\text{Ni}(\text{OH})_2$ and LiOH at a molar ratio of 1:1.05 at 600 °C for 6 hours under an oxygen flow, a procedure feasible for most materials laboratories. It is important to note that the 13:87 rocksalt-to-layered phase ratio is not a synthesis condition but rather a measure of quality control using quantitative SXR analysis after synthesis. As we indicated in our response to Reviewer #3, we agree that crucibles with varying temperature calibrations and gas flow schemes may affect growth kinetics. Therefore, the nominal heating temperature and time required to achieve the dual-phase composite with optimized electrochemical properties may vary across different laboratories. To address this, we identified the 13:87 rocksalt-to-layered phase ratio as a benchmark for an optimized structural configuration, independent of specific heating temperature and duration. The label "600C-6h" should be regarded as a sample designation, reflecting the experimental conditions specific to the heating devices used in our laboratories.

Reviewer #3's original comment pertained to the growth kinetics of intermediate phases and the ambient stability of the dual-phase composite. As shown in Figures 2a and 2b of our manuscript, due to sluggish Li-ion transport, the rocksalt-to-layered phase ratio evolves gradually even at 600 °C. For example, as indicated in Table S1, it takes 9 hours for the rocksalt fraction to decrease from 20.1% to 7.3%, demonstrating that the phase ratio can be precisely tuned by adjusting the sintering temperature and duration. Additionally, it is worth noting that the phase instability (or tunability) of the dual-phase composite occurs only at high temperatures. At room temperature, it exhibits stability comparable to most high-nickel NMC materials. The fact that the 600C-6h cathode maintains stable performance over 1,000 charge-discharge cycles while retaining its structure and morphology (Figure 2f and Figure S16) confirms its stability.

Instability at high temperatures is common in thermodynamically stable single-phase s-LNO as well. For example, single-phase s-LNO undergoes decomposition at 700 °C in less than 2 hours (J. Mater. Chem. A, 2020, 8, 1808–1820, Ref. 17 in our manuscript). Fine tuning the sintering temperature and duration is a standard approach for the sintering of layered materials.

The dual-phase composite d-LNO, like single-phase s-LNO and other high-nickel NMC materials, is highly sensitive to ambient environments and may react with moisture and carbon dioxide in the air. This sensitivity is a common feature of high-nickel NMC materials. Therefore, proper precautions in material storage, cell design, and preparation are essential to ensure consistent battery performance.

Finally, we would like to emphasize that the 13:87 ratio is not a sharp transition threshold that must be strictly maintained for the dual-phase composite to exhibit high performance. As shown in Figure 2b, when the rocksalt fraction varies between 7.3% and 20.1%, the capacity retention after 50 cycles remains above 95%, while the first discharge capacity remains around 195 mAh/g with a variation about 5%. Therefore, the synthesis conditions for our dual-phase cathode are reasonably forgiving and controllable for achieving the intended performance.

Reviewer #3 (Remarks to the Author):

I sincerely appreciate the authors' efforts in addressing my earlier comments and acknowledge the significant improvements in the manuscript. I believe the study presents valuable insights and merits consideration for publication.

However, I have reservations regarding the response to my earlier Comment 3, which addressed the influence of the rock-salt structure on diffusion kinetics. This aspect was not sufficiently addressed.

The authors convincingly present the idea of interlocking and controlling the growth of the layered structure from the rock-salt phase through a kinetically controlled process. The advanced characterization techniques used to demonstrate the phase composition—87% layered and 13% rock-salt.

Nonetheless, previous studies, including a highly influential work published by Prof. Clare P. Grey's group in Nature Materials (Xu, C., Märker, K., Lee, J. et al., 2021; <https://doi.org/10.1038/s41563-020-0767-8>), suggest that the rock-salt phase can restrict Li-ion mobility in Ni-rich layered cathodes. The findings in this manuscript appear contradictory to that established understanding.

It is crucial for the authors to provide a more detailed and comprehensive discussion addressing this apparent contradiction, including potential differences in experimental conditions or mechanistic interpretations. This will enhance the scientific rigor and clarity of the manuscript.

We sincerely appreciate the reviewer's insightful comments and the recommendation to consider Prof. P. Grey's work on cobalt-free high-Ni cathode materials. In our previous response, we noted that a higher fraction of the rock-salt (RS) phase leads to increased polarization at lower states of

charge (SOC) compared to single-phase LiNiO_2 (s-LNO). Initially, we attributed this to the electrochemical inactivity of the RS phase. However, as Grey et al. highlighted, it is also—or primarily—due to its poor Li-ion conductivity.

At higher SOC, while the RS phase retains its low Li-ion conductivity, its strong interlocking effect mitigates c-lattice shrinkage in the active layered phase during the H2–H3 transition, thereby enhancing lithium mobility. This dual effect explains why the rate capability comparison between the 600C-6h and s-LNO cathodes (**Figure S12**) shows no significant difference in the overall specific capacity dependence on rate.

Regarding the role of the RS phase, while both our study and Grey et al. (Nature Materials, 2021) discuss RS phase formation in Ni-rich layered oxides, the formation mechanisms, spatial distributions, and electrochemical impacts are fundamentally different. In our work, the RS phase is intentionally formed during synthesis via kinetic control, resulting in a bulk-integrated dual-phase composite with an interlocked RS-layered structure. In contrast, Grey et al. report an RS-like phase that forms progressively during electrochemical cycling, localizing at the surface of primary particles as a degradation product due to oxygen loss.

Another key difference is lattice matching: in our dual-phase composite, the RS phase exhibits minimal lattice mismatch with the layered phase across coherent grain boundaries formed during topotaxial growth. Conversely, Grey et al. observe significant lattice mismatch between the bulk layered structure and the surface RS-like phase, which they identify as a primary driver of structural degradation in their NMC electrode. In our case, the low mismatch indicates strong elastic coupling between the two phases, stabilizing the composite and enhancing cycling stability.

In summary, because the RS phases in the two studies differ in formation stage (synthesis vs. cycling), location (bulk vs. surface), and structural relationship with the layered phase, their roles in electrochemical performance are distinct. Experimentally, we see no contradiction between our findings and those of Grey et al.

As suggested by the reviewer, we have added the following discussion to the manuscript to address the apparent contradiction:

The effect of rocksalt domains within the d-LNO cathode on its electrochemical performance is fundamentally different from that of the NiO-like rocksalt phase formed on the surface of NMC cathodes during electrochemical cycling, as reported in the literature.^{37,38} The impact of the surface rocksalt phase on cathode performance is complex, often intertwined with the effects of the cathode/electrolyte interface (CEI) layer, and generally degrades cycling stability, especially at a high state of charge. In contrast, in our work, the rocksalt phase is intentionally formed during synthesis through kinetic control, resulting in a bulk-integrated dual-phase composite with an interlocked structure. This unique architecture suppresses lattice deformation during deep charging, thereby significantly enhancing the cycling stability of the d-LNO cathode.

With added references:

37. Tian, C.; Lin, F.; and Doeff M. M.; *Electrochemical Characteristics of Layered Transition Metal Oxide Cathode Materials for Lithium Ion Batteries: Surface, Bulk Behavior, and Thermal Properties*, *Acc. Chem. Res.* 2018, 51, 1, 89–96

38. Xu, C.; Marker, K.; Lee, J.; Mahadevegowda, A.; Reeves, P.; Day, M. F.; Groh, M.; Emge, S.; Ducati, C.; Mehdi, B. L.; Tang, C. C.; and Grey C. P.; *Bulk fatigue induced by surface reconstruction in layered Ni-rich cathodes for Li-ion batteries*, *Nature Materials* | VOL 20 | January 2021 | 84–92

That said, we would like to briefly comment on the "pinning" mechanism proposed by Grey et al. regarding RS-like surface reconstruction and its role in cathode degradation. They suggest that the RS phase mechanically "pins" the layered structure, preventing lattice contraction and thus inhibiting further delithiation. This assumption, however, contradicts established understandings of structural response to delithiation in Ni-rich cathodes.

Lattice contraction at high SOC is generally regarded as a consequence, not a prerequisite, of deep Li extraction [1–3]. In contrast, Grey et al. imply that preventing contraction leads to a "fatigued" phase, suggesting that contraction is essential for continued delithiation. However, they provide no theoretical or experimental evidence to support this claim.

Furthermore, the notion that a less contracted lattice hinders delithiation is inconsistent with literature. DFT calculations indicate that a slightly increased Li-slab thickness (i.e., a less contracted lattice) can significantly enhance Li mobility [1]. Based on this, various dopants have been introduced into layered cathodes to stabilize the lattice. For example, W-doping has been shown to reduce abrupt lattice contraction in LiNiO_2 , thereby improving its structural stability and Li mobility rather than impeding them [4].

If the RS phase indeed exerted strong mechanical constraint on the layered structure, it should undergo contraction itself. However, Grey et al. observe that the RS phase remains unchanged, undermining the idea of strong mechanical coupling. Moreover, the claim that the bulk layered structure within a primary particle conforms to a thin overlayer of the NiO-like phase contradicts common knowledge in the study of thin film epitaxy.

Epitaxial constraints and lattice contraction during charging are dynamic processes best studied through in situ characterization. Grey et al.'s XRD analysis does not detect the RS-like phase, and their ex situ STEM measurements only show lattice mismatch in a static state—insufficient evidence to support their conclusion.

We appreciate the opportunity to read this interesting paper. However, since the NiO-like surface phase discussed by Grey et al. does not directly relate to our work, we have chosen not to include this discussion in our manuscript.

References:

1. Kisuk Kang and Gerbrand Ceder, *Factors that affect Li mobility in layered lithium transition metal oxides*, *PHYSICAL REVIEW B* 74, 094105 (2006)
2. Hanlei Zhang, Hao Liu, Louis F. J. Piper, M. Stanley Whittingham and Guangwen Zhou, *Oxygen Loss in Layered Oxide Cathodes for Li-Ion Batteries: Mechanisms, Effects, and Mitigation*, *Chem. Rev.* 2022, 122, 6, 5641–5681

3. Wangda Li, Hooman Yaghoobnejad Asl, Qiang Xie, and Arumugam Manthiram, *Collapse of $\text{LiNi}_{1-x-y}\text{Co}_x\text{Mn}_y\text{O}_2$ Lattice at Deep Charge Irrespective of Nickel Content in Lithium-Ion Batteries*, *J. Am. Chem. Soc.* 2019, 141, 5097–5101
4. Hoon-Hee Ryu, Geon-Tae Park, Chong S. Yoon and Yang-Kook Sun, *Suppressing detrimental phase transitions via tungsten doping of LiNiO_2 cathode for next-generation lithium-ion batteries*, *J. Mater. Chem. A*, 2019, 7, 18580–18588